# Fast Data Attribution for Text-to-Image Models

**Sheng-Yu Wang**[1]   **Aaron Hertzmann**[2]   **Alexei A. Efros**[3]   **Richard Zhang**[2]   **Jun-Yan Zhu**[1]

[1]Carnegie Mellon University    [2]Adobe Research    [3]UC Berkeley

## Abstract

Data attribution for text-to-image models aims to identify the training images that most significantly influenced a generated output. Existing attribution methods involve considerable computational resources for each query, making them impractical for real-world applications. We propose a novel approach for scalable and efficient data attribution. Our key idea is to distill a slow, unlearning-based attribution method to a feature embedding space for efficient retrieval of highly influential training images. During deployment, combined with efficient indexing and search methods, our method successfully finds highly influential images without running expensive attribution algorithms. We show extensive results on both medium-scale models trained on MSCOCO and large-scale Stable Diffusion models trained on LAION, demonstrating that our method can achieve better or competitive performance in a few seconds, faster than existing methods by $2,500\times \sim 400,000\times$. Our work represents a meaningful step towards the large-scale application of data attribution methods on real-world models such as Stable Diffusion.

## 1   Introduction

Data attribution for text-to-image models [3, 4, 5, 6, 7, 8, 9, 10] aims to identify training images that most significantly influence generated images. For a given output, influence is defined counterfactually: *if influential images were removed from the training dataset, the model would lose its ability to generate the output.* However, directly identifying influential images by retraining models with all possible training subsets is computationally infeasible.

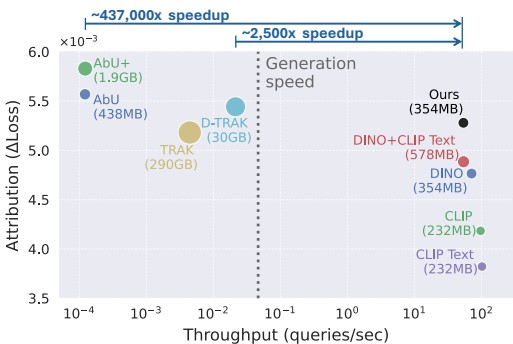

Figure 1: **Attribution performance vs. throughput.** Previous methods (AbU [1], D-TRAK [2]) offer high attribution performance but are computationally expensive for deployment. Fast image similarity using off-the-shelf features (DINO) lacks attribution accuracy. We distill slower attribution methods into a feature space that retains attribution performance while enabling fast deployment.

To make attribution more tractable, researchers have proposed several approaches to approximate this process. One common strategy is to estimate individual influence scores by examining the effect of removing single training images. Gradient-based methods, such as influence functions [11, 12, 13], approximate this removal process by calculating the inner product of the training and test images' model gradients, reweighted by the inverse Hessian. Unfortunately, computing and storing these gradients per training image is costly. Projecting gradients into lower dimensions reduces accuracy further [14], while still being computationally prohibitive for models with billions of parameters [15]. Another approach approximates influence by directly "unlearning" individual images, which improves accuracy but significantly increases runtime [1, 16, 17]. None of the above methods is practical for applications requiring fast test time performance. Furthermore,

39th Conference on Neural Information Processing Systems (NeurIPS 2025).

while text-to-image platforms typically charge users 5-10 cents per image [18, 19], existing attribution methods could cost way more per query due to heavier computation needs. This significant cost disparity, costing orders of magnitude more than generation revenue, prevents real-world deployment.

In this work, we propose a new method to overcome the tradeoff between computational complexity and attribution performance. Our key idea is to distill a slow, unlearning-based attribution method to a feature embedding space that enables efficient retrieval of highly influential training images. More concretely, leveraging off-the-shelf text and image encoders, we learn our embedding model's weights via learning-to-rank, supervised by the unlearning-based attribution method [1]. During deployment, our method scales to hundreds of millions of training images at a low cost.

However, this approach introduces several technical challenges. First, data curation is computationally intensive – collecting influence scores across large datasets (e.g., LAION-400M [20]) consumes significant GPU hours. To address this, we adopt a two-stage retrieval strategy: rapidly indexing [21] to select top candidates and then reranking them with more precise attribution methods. We show that our embedding function can be effectively learned from these reranked subsets. Second, designing objective functions to accurately predict dense rankings is non-trivial, as most learning-to-rank literature focuses on ranking fewer items. After evaluating several learning-to-rank objectives, we develop a simplified yet effective loss function.

We validate our method through extensive experiments. First, we benchmark our approach on MSCOCO using counterfactual evaluation, showing that our learned feature embeddings achieve strong attribution performance. We further compare runtime and storage costs against recent methods, as shown in Figure 1, demonstrating the best tradeoff between computational complexity and performance. Finally, we apply our approach to Stable Diffusion, achieving superior predictive performance on held-out test data. Our code, models, and datasets are at: `https://peterwang512.github.io/FastGDA`.

In summary, our contributions are:

- A new scalable approach to data attribution that employs a learning-to-rank method to learn features related to attribution tasks.
- A systematic study of key components that ensure efficient and effective learning-to-rank, including tailored objective functions and a two-stage data curation strategy.
- Extensive benchmarking against existing methods, demonstrating superior performance in both computational efficiency and attribution accuracy.
- The first successful application and evaluation of an attribution method on a large-scale model such as Stable Diffusion trained on LAION.

## 2   Related Works

**Data Attribution.** The interplay between training data and models has been extensively studied across various domains, including data selection [22, 23, 24, 25, 26, 27, 28, 29, 30, 31, 32], dataset mixing [33, 34, 35, 36, 37, 38], and data valuation [39, 40, 41, 42, 43, 44, 45, 46, 47]. In contrast, we focus on identifying training images influential to a given generated output in text-to-image models. For classification tasks, prior methods typically average contributions over models retrained on random subsets [48, 39, 40, 49, 46], inspired by Shapley values [50], or estimate gradient similarity across training checkpoints [51, 52].

Influence functions [12, 11] are also widely used, as they require no retraining or intermediate checkpoints. The main idea is to approximate changes in the model output loss after perturbing the training datapoint. This requires estimating gradients and an inverse Hessian matrix of the model parameters. To make evaluating the Hessian tractable, previous methods explore inverse Hessian-vector products [12], Arnoldi iteration [13], and Gauss-Newton approximation [14, 53, 54, 2, 15, 55], which is most commonly adopted for text-to-image models for its efficiency.

Despite these advances, challenges persist from high storage costs for model gradients across the dataset. To address this, researchers proposed several solutions, including estimating influence at test time [54, 1, 55, 16, 17] without storing gradients at a cost of increased runtime, and gradient dimensionality reduction through random projections [14, 53, 2] or low-rank adaptors [56, 15]. These methods introduce a tradeoff between computational resources (storage, runtime) and attribution

performance, as heavier approximations worsen attribution [2, 1]. To address this, we propose learning a feature space specifically trained to predict attribution results obtained by computationally expensive but accurate methods.

**Learning to Rank (LTR)** aims to predict item rankings [57] and are broadly categorized into pointwise, pairwise, and listwise approaches. Pointwise methods independently score each item, applying methods such as multiclass classification [58], ordinal loss [59], or regression [60]. Pairwise methods are trained to predict relative orderings between pairs, leveraging boosting [61], SVM [62], and neural networks with cross-entropy [63] or rank-based losses [64]. Listwise methods optimize the entire list directly [65, 66, 67, 68], aiming to improve ranking metrics such as NDCG [69], mAP, Precision@k, and Recall@k [70]. LTR benchmarks typically involve ranking fewer than 1,000 items [71, 72]. In contrast, our task involves ranking significantly larger lists (e.g., 10,000 items). For simplicity, we focus on a pointwise method and propose a simple, effective variant that outperforms strong baselines in our application.

**Representation Learning.** The representations learned in deep networks [73, 74, 75] often translate across tasks [76, 77, 78]. For example, a representation learned from an image classification dataset [79] can be efficiently repurposed for tasks, from detecting objects [76] to modeling human perception [80, 81]. Unsupervised [82, 83] or self-supervised learning [84, 85, 86, 87, 88] methods aim to learn image representations without text labels. A particularly effective training objective is the contrastive loss [89, 90, 91, 92, 93], where co-occurring data or two views of the same datapoint (such as text and image in CLIP [94]), are associated together, in contrast to other unrelated data. Previous work by Wang et al. [95] evaluate and tune a representation towards attribution in the customization setting. In our work, we leverage counterfactually predictive unlearning methods and learn representations for general-purpose attribution.

# 3 Method

Our goal is to learn an attribution-specific feature using a collection of attribution examples, generated from an accurate but slow attribution method. Section 3.1 overviews the attribution algorithm we adopt and explains our data collection process. In Section 3.2, we introduce our learning to rank objective for training attribution-specific features.

## 3.1 Collecting Attribution Data

Following the convention from Wang et al. [1], a text-to-image model $\theta_0 = \mathcal{A}(\mathcal{D})$ is trained with learning algorithm $\mathcal{A}$ on dataset $\mathcal{D} = (\mathbf{x}_i, \mathbf{c}_i)_{i=1}^N$, where $\mathbf{x}$ and $\mathbf{c}$ denote an image and its conditioning text, respectively. The model $\theta_0$ generates a synthesized image $\hat{\mathbf{x}}$ conditioned on caption $\mathbf{c}$. To simplify notation, we denote a synthesized pair as $\hat{\mathbf{z}} = (\hat{\mathbf{x}}, \mathbf{c})$ and a training pair as $\mathbf{z}_i = (\mathbf{x}_i, \mathbf{c}_i)$.

Data attribution aims to attribute the generation $\hat{\mathbf{z}}$ to its influential training data. We define a data attribution algorithm $\tau$, which assigns an attribution score $\tau(\hat{\mathbf{z}}, \mathbf{z}_i)$ to each training datapoint $\mathbf{z}_i$, with higher score indicating higher influence. We assume that $\tau$ has access to all training data, parameters, and the loss function, but omit them for notational brevity.

**Attribution by Unlearning.** We collect attribution examples using Attribution by Unlearning (AbU) [1], one of the leading attribution methods for text-to-image models, shown in the top of Figure 2. The method runs machine unlearning [96, 97, 98, 99] on the synthesized image and evaluates which of the training images are forgotten by proxy. The authors observe that the forgotten training images are more likely to be influential. Intuitively, this reverses the question: which training points must the model forget to unlearn the synthesized image?

Formally, to unlearn the synthesized image and evaluate which training images are "forgotten", we start with the pre-trained model $\theta_0$, and apply certified unlearning [96] on the synthesized sample $\hat{\mathbf{z}}$:

$$\theta_{-\hat{\mathbf{z}}} = \theta_0 + \frac{\alpha}{N} F^{-1} \nabla \mathcal{L}(\hat{\mathbf{z}}, \theta), \tag{1}$$

this creates unlearned model $\theta_{-\hat{\mathbf{z}}}$, where $\alpha$ is the step size, $N$ is the size of the training set, and $F$ is the Fisher information matrix. $\mathcal{L}$ is the training loss of text-to-image diffusion model, as defined in Appendix A.1. This constitutes a one-step Newton update to maximize the loss of the synthesized

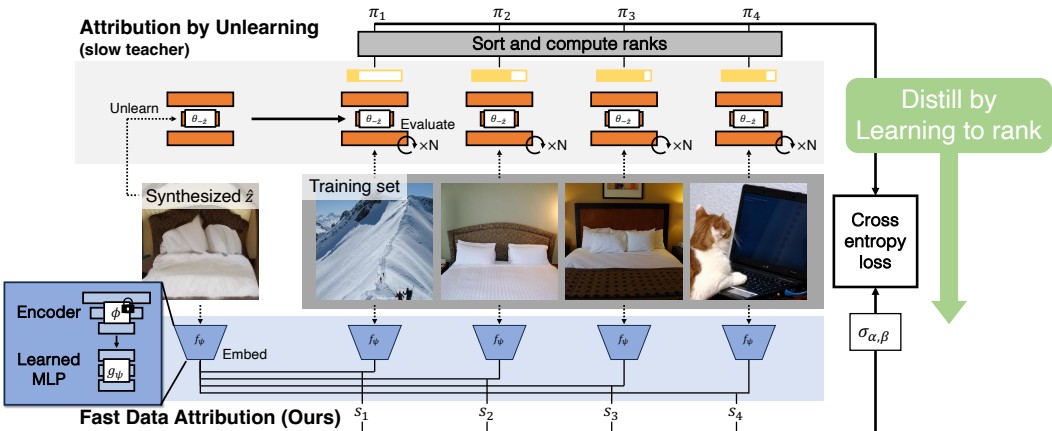

Figure 2: **Our method.** Given a synthesized sample, data attribution aims to find which elements in the training set are more influential. (Top) Attribution by Unlearning (AbU) is a slow but accurate method. It works by unlearning a synthesized example and evaluating the change in reconstruction loss on each training image, where each evaluation takes many forward passes. We generate AbU scores, using them to train an attribution-focused embedding (bottom), so that attribution can be performed by fast similarity search, while retaining the accuracy of the slower AbU method.

sample through gradient ascent, regularized by elastic weight consolidation (EWC) loss [100] to retain other knowledge.

After unlearning, the attribution score is assigned to each training sample $\mathbf{z}$ by evaluating the training loss change, before and after unlearning:

$$\tau(\hat{\mathbf{z}}, \mathbf{z}) = \mathcal{L}(\mathbf{z}, \theta_{-\hat{\mathbf{z}}}) - \mathcal{L}(\mathbf{z}, \theta_0). \tag{2}$$

The loss change in $\tau(\hat{\mathbf{z}}, \mathbf{z})$ measures how much the unlearned model loses its capability to represent each training image. The method is effective but computationally expensive, as it requires evaluating the unlearned model on every training image at runtime.

**Attribution by Unlearning+.** The Fisher information matrix of a deep network is difficult to compute and store, as its size grows quadratically with the number of model parameters. In AbU, Wang et al. [1] simply approximate this using a diagonal approximation. We find that replacing the diagonal approximation with the Eigenvalue-corrected Kronecker Factorization (EK-FAC) approximation [101] yields better performance and refer to this as **AbU+**.

While this method is accurate, it takes a significant amount of runtime. For example, it takes 2 hours to process a single query for a training dataset with 100K images. We aim to distill this slow but accurate method into a fast embedding next.

**Two-stage data collection.** Recall that our goal is to collect a dataset of attribution scores between synthesized and training examples. However, the expensive runtime prohibits collecting data across all training samples. We resolve this using a two-stage data collection process to improve efficiency. We note that most training samples do not meaningfully contribute to a synthesized example, so evaluating *all* training samples $\mathcal{D}$ for a given query is unnecessary. To narrow the search space, we first obtain the $K$ nearest neighbors of the synthesized sample using off-the-shelf features $\mathcal{D}_{\hat{\mathbf{z}}} \subset \mathcal{D}$. We then collect attribution scores for them $\mathcal{S}_{\hat{\mathbf{z}}} = \{\tau(\hat{\mathbf{z}}, \mathbf{z}_k)\}_{\mathbf{z}_k \in \mathcal{D}_{\hat{\mathbf{z}}}}$. As long as the subset contains the most influential images, the data is suitable for learning to identify them. Guo et al. [102] also explored a similar two-stage approach, but to reduce runtime cost for attribution, whereas we apply it to speed up data collection.

Our final dataset contains query images, corresponding training subset, and collected attribution scores: $\hat{\mathbf{z}}, \mathcal{D}_{\hat{\mathbf{z}}}, \mathcal{S}_{\hat{\mathbf{z}}}$.

## 3.2 Learning to Rank From Attribution Data

Given query $\hat{\mathbf{z}}$ and attribution scores $\mathcal{S}_{\hat{\mathbf{z}}}$ in dataset $\mathcal{D}_{\hat{\mathbf{z}}}$, we obtain normalized ranked scores $\pi_{\hat{\mathbf{z}}}^i \in [\frac{1}{K}, \frac{2}{K}, ..., 1]$ for each training sample $\mathbf{z}_i$, where the most influential sample is assigned $\frac{1}{K}$, and the least assigned 1.

**Predicting attribution rank.** We then learn a function $r_\psi(\hat{\mathbf{z}}, \mathbf{z}_i)$ to efficiently predict the true rank $\pi_{\hat{\mathbf{z}}}^i$ of the sample, as shown in the bottom branch of Figure 2. We parameterize $r_\psi(\hat{\mathbf{z}}, \mathbf{z}_i) = \cos(f_\psi(\hat{\mathbf{z}}), f_\psi(\mathbf{z}_i))$ to compare cosine similarity across feature embeddings. This enables one to store embeddings $f(\mathbf{z}_i)$ and perform simple feature similarity search at attribution time.

We parameterize feature embedding $f_\psi = g_\psi \circ \phi$, where $\phi$ is a pretrained network and $g_\psi$ is a learned MLP that maps pretrained features towards an embedding focused on attribution.

**Learning to rank by cross-entropy.** We adopt the pointwise learning-to-rank approach [58, 59, 60] and apply cross-entropy loss to optimize each rank prediction individually. We then learn $r_\psi$ through binary cross-entropy loss $\ell_{\text{BCE}}$:

$$\mathcal{L}(\psi, \alpha, \beta) = \mathop{\mathbb{E}}_{\hat{\mathbf{z}} \sim \hat{\mathcal{Z}}, \mathbf{z}_i \sim \mathcal{D}_{\hat{\mathbf{z}}}} \ell_{\text{BCE}}\Big( \pi_{\hat{\mathbf{z}}}^i, \sigma_{\alpha,\beta}\big( r_\psi(\hat{\mathbf{z}}, \mathbf{z}_i) \big) \Big), \tag{3}$$

where we randomly sample synthesized images from a collection $\hat{\mathcal{Z}}$ and training images $\mathcal{D}_{\hat{\mathbf{z}}}$, and function $\sigma_{\alpha,\beta}(x) = \frac{1}{1+e^{-(\alpha x + \beta)}}$ is a logit with a learned affine scaling. We also explored other alternatives. Direct regression [60, 103] performs poorly, whereas Ordinal regression loss from Crammer and Singer [59] delivers competitive ranking performance. However, implementing the loss requires changes in our rank function designs, such that it no longer supports fast feature similarity search at inference time. A full derivation of the ordinal loss, along with more details of the alternative ranking losses, could be found in the supplementary material.

**Sampling strategies.** To further improve accuracy and reduce attribution-score computation costs, we modify our sampling of training examples in two ways. First, sampling only from top-$K$ neighbors focuses the model on fine-grained distinctions but hurts its ability to recognize unrelated images. To counteract this, with probability 0.1, we draw a random example $\mathbf{z}$ *outside* the neighbor set and assign it the worst rank: $\pi_{\hat{\mathbf{z}}}^i = 1$. This encourages the model to rank non-neighbor images last.

Second, computing attribution scores on all $K$ neighbors at every step still incurs $\mathcal{O}(K)$ cost. We therefore subsample $M < K$ candidates uniformly at each iteration, compute true ranks only on that subset, and train using those $M$ examples. In practice, we find that choosing $M \approx 0.1K$ (or even smaller) preserves ranking performance, while greatly reducing per-query curation time. We provide a detailed study on how each strategy affects accuracy and efficiency in Section 4.1.

## 4 Experiments

We address two questions in this section: (Q1) *Rank prediction*: Can our tuned feature improve attribution-rank accuracy? (Q2) *Counterfactual forgetting*: Does better rank prediction translate into stronger counterfactual prediction, the "gold standard" to evaluate attribution? This counterfactual prediction verifies whether the model forgets about the synthesized sample when its influential data are removed. We evaluate Q1 on both MSCOCO and Stable Diffusion models, and Q2 on MSCOCO only, since repeated retraining on Stable Diffusion is computationally prohibitive.

### 4.1 MS-COCO models

We follow previous testbeds, using the same latent diffusion model [4] trained on the 100k MSCOCO dataset [104]. The model is the same as used in AbU [1] and Georgiev *et al.* [53], and we use the test set from AbU, which contains 110 synthesized queries.

**Rank prediction metrics (Q1).** We collect ground-truth ranks with AbU+ on the 110 queries, where we rank every training data point. To train our model, we generate 5000 images, using other prompts in MSCOCO. To build our dataset, for each query, we select the top 10k nearest neighbor candidates in the DINO feature space and rank candidates with AbU+, totaling 50M attribution ranks. We take 4900 queries for training and 100 for validation. We measure rank-prediction accuracy on the 110-query test set by mean average precision (mAP) in a binary-retrieval setup: the top $L$

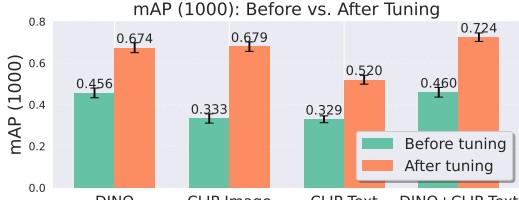 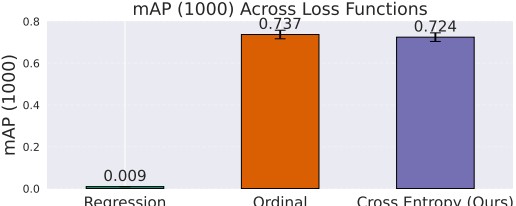

Figure 3: (Left) **Feature spaces.** We compare different feature spaces, before and after tuning for attribution. We measure mAP to the ground truth ranking, generated by AbU+. While text-only embeddings perform well before tuning, image-only embeddings become stronger after tuning. Combining both performs best and is our final method. (Right) **Ranking loss functions.** Simple MSE regression does not converge well. Ordinal loss works well, but does not support fast similarity search at inference time. We use cross-entropy, which achieves performance similar to ordinal loss while supporting similarity search. We report 1-standard error in the plots.

training points from the ground truth rank are labeled positive. We denote this as **mAP** ($L$) for $L \in \{500, 1000, 4000\}$. We note that this metric is different from conventional mAP@$K$, where AP is evaluated for the first $K$ data points only. Instead, we use $L$ only to define positives while evaluating over the full training set. We also report other ranking metrics in Appendix C.2.

**Counterfactual forgetting metrics (Q2).** Following AbU, for each test query, we remove the top $k$ influential images ($k = 500, 1000, 4000$), retrain the model *from scratch*, and measure how much the retrained model forgets via:

- **Loss change** $\Delta\mathcal{L}(\hat{\mathbf{z}}, \theta)$: the increase in query $\hat{\mathbf{z}}$ loss under the retrained model relative to the original pretrained model.

- **Generation deviation** $\Delta G_\theta(\epsilon, \mathbf{c})$: the difference between the original and re-generated image (using the same noise seed $\epsilon$), where larger deviation indicates stronger forgetting [53]. As in AbU, we report the difference in mean square error (MSE) and CLIP similarity.

Note that this test is expensive, as it involves retraining a model for each combination of synthesized image and attribution method, but is a gold-standard counterfactual test for attribution. We select and study various design choices of our ranking model using the inexpensive rank-prediction metrics, then apply the much costly counterfactual forgetting evaluation to confirm that the tuned feature space indeed improves attribution.

**Which feature space to tune?** Figure 3 (left) reports the effect of tuning different feature spaces. Before tuning, using a text embedding (CLIP-text) beats image-only embeddings (DINO or CLIP). However, after tuning, the image-only embeddings achieve higher performance than text-only. The best results come from concatenating DINO and CLIP-Text: after tuning this combined feature, we achieve the highest prediction accuracy, harnessing both visual and text signals.

**Learning-to-rank objectives.** Figure 3 (right) compares three objective functions. We first implement a simple mean-squared-error (MSE) loss to regress ground-truth ranks, as used in previous work [60, 103]. However, this formulation fails to converge and yields poor retrieval accuracy.

Next, we find that ordinal loss [59] delivers competitive ranking performance. However, as mentioned in Section 3.2, this loss function is incompatible with fast feature search at inference time. Finally, we adopt a cross-entropy loss on rank labels (normalized by $K$), which preserves the efficient cosine-similarity search, while matching the ordinal loss in mAP($L$) across all thresholds $L$. Please see additional comparisons in Appendix C.2.

**Data scaling.** Figure 4 (left) shows how rank-prediction accuracy improves as we increase the number of training samples. We see steep gains when moving from small to moderate dataset sizes, followed by diminishing returns as size grows further. This "elbow" behavior suggests that a few thousand attribution query examples suffice to capture the bulk of the ranking signal.

**Sampling images outside the neighbor set.** Figure 4 (right) explores the impact of injecting random "negative" samples—images not among the top-$K$ neighbors—during training. A modest 10% sampling rate of non-neighbors raises accuracy from 0.709 to 0.724, but higher rates steadily degrade performance. This pattern suggests that occasional negatives help the model maintain global

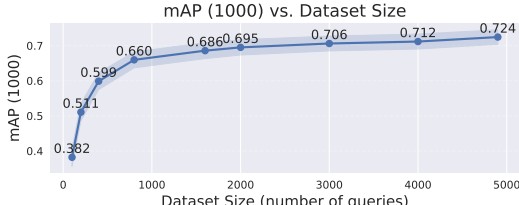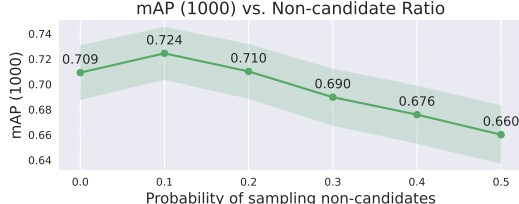

Figure 4: (Left) **Data scaling.** We investigate the impact of the number of synthesized queries. Note that each synthesized query contains attribution scores with 10k training points. We find that the performance quickly improves and saturates. (Right) **Sampling outside the neighbor set.** We vary the probability of selecting non-nearest neighbor images when building the attribution dataset. Using a few randomly sampled, unrelated images from the training set helps keep the learned attribution model, while having too many impedes the learning. We report 1-standard error in the plots.

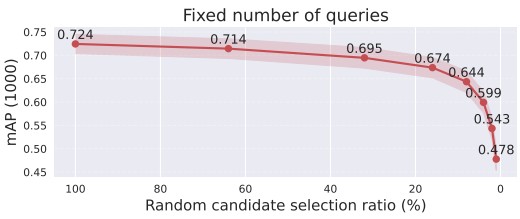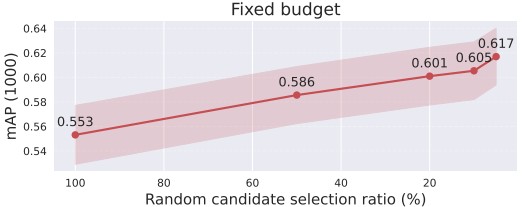

Figure 5: **Sampling strategies of dataset construction.** (Left) For each query, we randomly select from the 10k neighbors to learn from. Reliable rankings can be learned, even with relatively fewer training images per query are provided in the dataset. (Right) Given a fixed budget of 2.45M query-training attribution ranks, we test trading off between fewer training images per query and more synthesized queries. We find that at this budget, more query images with fewer training images are beneficial. We report 1-standard error in the plots.

distances (assigning truly unrelated images worse ranks), yet over-sampling them distracts from the core task of fine-grained ranking among the most relevant candidates.

**Random sampling of nearest-neighbor candidates.** Figure 5 (left) fixes the number of queries and varies the fraction of top-$K$ candidates used for training. We observe that rank-prediction accuracy drops mildly until the candidate subset falls below 20%, indicating that reliable relative rankings can be learned from a small fraction of neighbors. Figure 5 (right) explores a fixed compute-budget regime (constant total queries $\times$ selected candidates) of 2.45M query-training attribution ranks. (Recall our complete dataset has 50M). In this setting, one can reduce the number of training images sampled per query and increase the number of distinct queries. We find that this boosts accuracy. This trade-off suggests that under budget constraints, it is better to sample fewer candidates per query to evaluate more queries. Accordingly, we adopt a 20% neighbor-sampling rate for collecting attribution examples in our Stable Diffusion experiments (Section 4.2).

**Runtime vs. counterfactual performance.** In Table 1, we compare performance and run-time against several baselines. Runtime and storage are shown on the left. We gather runtime values for each method by warmstarting 10 queries (if applicable) and then averaging over 20 queries. We run on a single Nvidia A100 80GB for benchmarking. The right columns of Table 1 show the counterfactual performance – retraining a model from scratch without a subset of identified images from an attribution method – and seeing if the targeted synthesized image is damaged in the subsequent model (Loss deviation) or generation (Image deviation).

The influence and unlearning-based methods offer tradeoffs between latency, performance, and storage. Unlearning-based AbU, and our improved AbU+ variant achieve the highest attribution performance. However, the method requires hours to run, due to repeated function evaluations on each training image. While the influence function-based methods are faster, they are lower performing, with D-TRAK performing relatively well at fast inference times (46.7 sec). However, note that this is still longer than the time to generate an image (21.5 sec for 50-DDIM steps), and the method requires storage of the preconditioned gradients on the training set (30GB), which is larger than the images in the training set itself (19 GB).

| Family | Method | Efficiency | | Loss deviation $\Delta\mathcal{L}(\hat{\mathbf{z}},\theta)$ $\uparrow$ ($\times10^{-3}$) | | | Image deviation $\Delta G_\theta(\epsilon,\mathbf{c})$ MSE $\uparrow$ ($\times10^{-2}$) | | | CLIP $\downarrow$ ($\times10^{-1}$) | | |
|---|---|---|---|---|---|---|---|---|---|---|---|---|
| | | Latency | Storage | 500 | 1000 | 4000 | 500 | 1000 | 4000 | 500 | 1000 | 4000 |
| Random | Random | – | – | 3.51±0.03 | 3.46±0.03 | 3.47±0.03 | 4.09±0.06 | 4.07±0.06 | 4.05±0.06 | 7.86±0.03 | 7.85±0.03 | 7.85±0.03 |
| Influence | TRAK | 3.76 min | 290 GB | 5.18±0.14 | 5.77±0.16 | 7.05±0.16 | 4.67±0.21 | 4.68±0.24 | 4.75±0.20 | 7.65±0.09 | 7.63±0.09 | 7.49±0.09 |
| | JourneyTRAK | 3.64 min | 290 GB | 4.44±0.11 | 4.87±0.13 | 5.72±0.15 | 4.77±0.19 | 5.36±0.23 | 5.42±0.24 | 7.68±0.09 | 7.53±0.09 | 7.53±0.09 |
| | D-TRAK | 46.7 sec | 30 GB | 5.44±0.16 | 6.60±0.22 | 9.59±0.33 | **5.86±0.24** | **6.43±0.25** | **7.82±0.30** | 7.31±0.10 | 7.06±0.09 | 6.44±0.09 |
| Unlearn. | AbU | 2.28 hr | 438 MB | 5.57±0.16 | 6.75±0.22 | 9.78±0.32 | 5.07±0.21 | 5.69±0.24 | 6.07±0.22 | 7.35±0.09 | 7.00±0.09 | 6.36±0.10 |
| | AbU+ | 2.28 hr | 1.9 GB | **5.83±0.17** | **7.13±0.22** | **10.70±0.32** | 5.64±0.25 | 6.20±0.24 | 7.54±0.25 | **7.15±0.10** | **6.83±0.10** | **5.80±0.09** |
| Text | CLIP$_{\text{Text}}$ | 9.8 ms | 232 MB | 3.82±0.12 | 4.19±0.14 | 5.52±0.25 | 4.12±0.20 | 4.30±0.19 | 4.56±0.19 | 7.84±0.08 | 7.72±0.09 | 7.41±0.08 |
| Image | Pixel | 603.6 ms | 19 GB | 3.59±0.10 | 3.64±0.10 | 3.98±0.11 | 4.34±0.19 | 4.30±0.21 | 4.90±0.21 | 7.85±0.10 | 7.80±0.09 | 7.69±0.10 |
| | CLIP | 10.3 ms | 232 MB | 4.18±0.14 | 4.69±0.18 | 6.44±0.32 | 4.35±0.20 | 4.57±0.21 | 5.20±0.22 | 7.63±0.09 | 7.54±0.08 | 6.79±0.08 |
| | DINO | 11.6 ms | 354 MB | 4.76±0.15 | 5.60±0.20 | 8.06±0.35 | 4.51±0.16 | 5.29±0.22 | 5.88±0.21 | 7.41±0.09 | 7.10±0.09 | 6.27±0.10 |
| | DINOv2 | 27.9 ms | 464 MB | 3.89±0.12 | 4.29±0.15 | 6.26±0.33 | 4.30±0.20 | 4.59±0.20 | 5.09±0.19 | 7.68±0.09 | 7.66±0.08 | 6.98±0.09 |
| | AbC (CLIP) | 10.4 ms | 232 MB | 4.35±0.13 | 4.92±0.17 | 6.94±0.32 | 4.55±0.20 | 5.05±0.22 | 5.63±0.23 | 7.52±0.09 | 7.26±0.09 | 6.54±0.09 |
| | AbC (DINO) | 11.7 ms | 354 MB | 4.75±0.15 | 5.53±0.20 | 8.11±0.35 | **4.78±0.22** | 4.95±0.20 | 5.81±0.21 | 7.51±0.09 | 7.18±0.09 | 6.29±0.09 |
| Im.+Text | DINO+CLIP$_{\text{Text}}$ | 18.6 ms | 578 MB | 4.88±0.16 | 5.55±0.20 | 8.02±0.34 | 4.63±0.20 | **5.30±0.23** | 5.83±0.22 | 7.42±0.09 | 7.15±0.10 | 6.29±0.10 |
| Ours | DINO+CLIP$_{\text{Text}}$ (Tuned) | 18.7 ms | 354 MB | **5.28±0.17** | **6.44±0.24** | **9.35±0.35** | **4.78±0.22** | **5.30±0.22** | **6.27±0.24** | **7.37±0.09** | **7.05±0.09** | **6.05±0.09** |

Table 1: **Runtime and counterfactual leave-$K$-out evaluations.** We show the runtime and storage requirements of different attribution methods. Note that in this setting, an image takes 21.5 seconds to generate. Influence and unlearning-based methods are slower than generation, highlighted in yellow and red, while search-based embedding methods are faster than generation, shown in green. We show storage costs for attribution. Methods are colored relative to the storage cost of the training set (19 GB), more than the training set as red, within $10\times$ as yellow, and significantly less as green. Following the "gold-standard" metrics from Wang et al. [1], we measure the ability of different attribution methods to predict counterfactually significant training images. That is given a synthesized image $\hat{\mathbf{z}}$, we train leave-$K$-out models for each of the attribution methods and track $\Delta\mathcal{L}(\hat{\mathbf{z}},\theta)$, the increase in loss change, and $\Delta G_\theta(\epsilon,\mathbf{c})$, deviation of generation. We report results over 110 samples, and gray shows the standard error. Across each efficiency regime (slower or faster than generation time), we **bold** the best method and underline methods that are within a standard error. Of the fast methods, our method performs the best at attribution.

On the other hand, embedding-based methods offer low-storage (100s of megabytes), depending on the size of the embedding, and fast run-times (tens of milliseconds). As such, these methods can be tractable, as they are cheap to create relative to the generation itself. As the models we test are learning a mapping from text to image, we explore both text and image-based descriptors. As a baseline, we test if a text descriptor is sufficient for attribution, using the CLIP text encoder. While it performs better than random, text-only is not sufficient, and the visual content is indicative of which training images were used. While one can use an off-the-shelf visual embedding and achieve visually similar images from the training set, they are not guaranteed to be counterfactually predictive and appropriate for attribution either. Previous work [1] found DINO to be a strong starting point.

From the ranking evaluation above, for our method, we select the best-performing model, which is tuned DINO+CLIP Text feature with 0.1 probability sampling images outside the neighbor set, using all the data within the neighbor set. In Figure 1, we plot the attribution performance (loss change with $k = 500$) vs. throughput (reciprocal of run-time). Our method achieves strong attribution performance overall, comparable to influence-function-based methods, at orders of magnitude higher speed. We show qualitative results in Figure 7 (left) and provide more results in Appendix C.2.

## 4.2 Stable Diffusion

We use Stable Diffusion v1.4 [4, 105] for our experiments. We collect generated images and captions from DiffusionDB [106] as attribution queries. Since retraining is prohibitive for such large models, we only report rank prediction metrics in this setting. In Figure 7 (right), we show a qualitative example of DINO + CLIP Text feature before and after tuning.

**Rank prediction metrics (Q1).** Same as the setup in MSCOCO (Section 4.1), we collect ground truth queries and collect attribution examples of them using AbU+, where the queries are split for training, validation, and testing. Different from MSCOCO, it is prohibitively costly to run AbU+ over the entire dataset (LAION-400M [20]). Hence, for each test query, we retrieve 100k nearest neighbor candidates from LAION-400M through CLIP index [21] and obtain ground truth attribution ranks for those candidates. For training and validation, following studies in Section 4.1, for each query, we retrieve 100k nearest neighbors. To collect more diverse queries within a fixed budget, we only rank a 20% random selection of them for supervision. We collect 5000 queries for training and 50 queries for validation, for a total of 101M query-training attribution ranks. We report mAP ($L$)

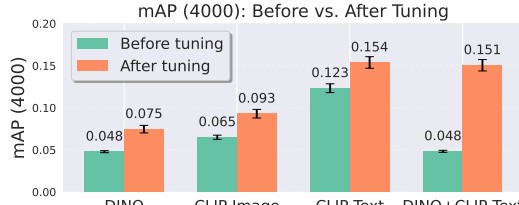 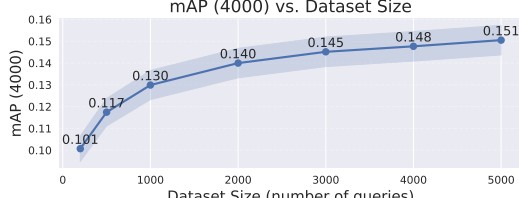

Figure 6: **Stable Diffusion ranking results.** (Left) **Tuning performance**. We see similar trends as with MS-COCO, with strongest performing embedding using both text and image features. (Right) **Data scaling**. Performance increases with query images, increasing additional gains with more compute dedicated to gathering attribution training data. We report 1-standard error in the plots.

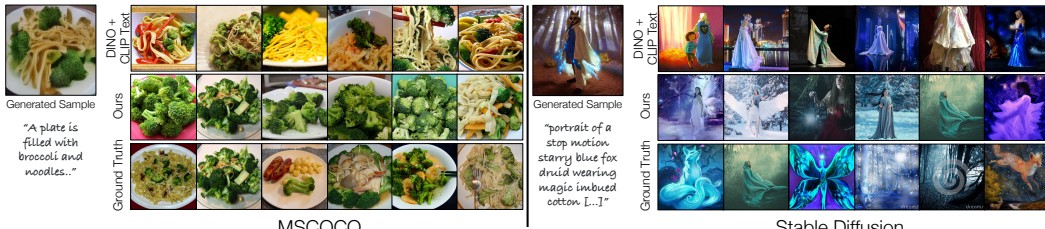

Figure 7: **Qualitative examples on MSCOCO (left) and Stable Diffusion (right).** For each generated image and its text prompt on the left, we show top-5 training images retrieved by: *DINO + CLIP-Text* (top row), *Ours* (middle row), and the ground-truth influential examples via AbU+ (bottom row). Compared to the untuned baseline, our distilled feature space yields attributions that match the ground-truth examples more closely.

for $L \in \{500, 1000, 4000\}$, where we use the tuned features to rank the 100k candidates and check whether this prediction aligns with the ground truth.

**Which feature space to use?** Figure 6 (left) reports performance of different features before and after tuning. Similar to the findings in MSCOCO models, tuning feature spaces consistently improves the prediction results. However, in contrast to MSCOCO models, where image feature is an important factor for accurate rank prediction, we observe the opposite in Stable Diffusion. In fact, the results indicate that text features are strictly necessary to yield good tuning performance. Using DINO and CLIP-Image features significantly underperforms the ones with text features. This indicates that AbU+ tends to assign attribution scores that are more correlated with text feature similarities. We discuss this more in Appendix C.2

**Data scaling.** Figure 6 (right) reports performance improvements with respect to dataset size. Similar to the findings in MSCOCO models, there are steep gains from small to moderate dataset sizes, and the rate of gain decreases as size grows further. However, we note that even with the full dataset at 5000 queries (100M data points), the performance increase has not saturated. With more compute, one could collect more data to improve ranking performance.

## 5 Discussion, Broader Impacts, and Limitations

Data attribution is a quest to understand model behavior from a data-centric perspective. It can potentially aid practical applications such as compensation models, which could help address the timely issue surrounding the authorship of generative content [107, 108, 109, 110]. Our method reduces the runtime and storage cost of the data attribution algorithm, which is a step towards making data attribution a feasible solution for a compensation model.

While our work demonstrates a good tradeoff between compute resources and attribution performance, there are additional avenues for future work. First, the learning to rank approach does not distill the raw attribution score, just the relative ranking in the training set, so the degree of influence and how diffuse or concentrated the influence may be lost as well. Further exploring and characterizing the degree of influence is an area of future work. Secondly, as our method is distilled from a teacher method, failure modes will also be inherited. However, in this work, we have shown that future broader improvements in attribution methods can benefit a faster method through distillation. Besides developing a fast

attribution method for diffusion models, there are opportunities for applying attribution to other widely used models (e.g., flow matching [111, 112], one-step models [113, 114, 115, 116]) and making attribution more explainable to the end users.

**Acknowledgments.** We thank Simon Niklaus for the help on the LAION image retrieval. We thank Ruihan Gao, Maxwell Jones, and Gaurav Parmar for helpful discussions and feedback on drafts. Sheng-Yu Wang is supported by the Google PhD Fellowship. The project was partly supported by Adobe Inc., the Packard Fellowship, the IITP grant funded by the Korean Government (MSIT) (No. RS-2024-00457882, National AI Research Lab Project), NSF IIS-2239076, and NSF ISS-2403303.

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

## A    Formulations

Here we expand the main paper's formulations with more details.

### A.1    Diffusion Models.

Section 3 of the main paper describes our formulation, which uses the loss $\mathcal{L}$ of the generative model being used. Our experiments are on diffusion models, and we describe the loss term here. Diffusion models [117, 118, 119] learns to reverse a data noising process, defined as $\mathbf{x}_0, \ldots, \mathbf{x}_T$, where a clean image $\mathbf{x}_0$ is gradually diffused to a pure Gaussian noise $\mathbf{x}_T$ over $T$ timesteps. At timestep $t \in [0, T]$, noise $\epsilon \sim \mathcal{N}(0, I)$ is blended into the clean image $\mathbf{x}_0$ to form a noisy image $\mathbf{x}_t = \sqrt{\bar{\alpha}_t}\mathbf{x}_0 + \sqrt{1 - \bar{\alpha}_t}\epsilon$, where $\bar{\alpha}_t$ defines the noise schedule. The training loss of the diffusion models is to encourage accurate denoising of $\mathbf{x}_t$ by predicting the noise:

$$\mathbb{E}_{\epsilon, \mathbf{x}, \mathbf{c}, t} \left[ w_t \|\epsilon - \epsilon_\theta(\mathbf{x}_t, \mathbf{c}, t)\|^2 \right], \tag{4}$$

where $w_t$ is a weighting term typically set to 1 [117], $\epsilon_\theta$ is the denoising model, and $\mathbf{c}$ is the text condition. At inference, the denosing model $\epsilon_\theta$ takes in random Gaussian noise and gradually denoises it to the learned data distribution.

Evaluating the diffusion loss in Equation 4 requires a Monte-Carlo estimate, and we estimate it by taking averages over different noise samples and timesteps. Section B provides implementation details of the Monte-Carlo estimate.

### A.2    Influence function.

In the main text, we discussed that influence function methods suffer from a tradeoff between computation cost and attribution performance. To understand this, we briefly recap the influence function from Koh and Liang [12]. Let

$$\mathcal{R}(\theta) = \frac{1}{N} \sum_{i=1}^{N} \mathcal{L}(\mathbf{z}_i, \theta), \quad \theta_0 = \arg\min_\theta \mathcal{R}(\theta), \quad H = \nabla_\theta^2 \mathcal{R}(\theta_0), \tag{5}$$

where $\mathcal{R}$ is the full training loss, $\{\mathbf{z}_i\}$ are the training examples, and $H$ is the Hessian of the training loss. We estimate the effect of removing a single training point $\mathbf{z}$ via a small perturbation $\epsilon$:

$$\theta_{-\mathbf{z},\epsilon} = \arg\min_\theta \{\mathcal{R}(\theta) - \epsilon \mathcal{L}(\mathbf{z}, \theta)\}, \tag{6}$$

where $\theta_{-\mathbf{z},\epsilon}$ is the optimal model under the perturbed loss. $\theta_{-\mathbf{z},\epsilon}$ statistfies the stationary condition:

$$0 = \nabla_\theta \mathcal{R}(\theta_{-\mathbf{z},\epsilon}) - \epsilon \nabla_\theta \mathcal{L}(\mathbf{z}, \theta_{-\mathbf{z},\epsilon}). \tag{7}$$

From Equation 7, taking a Taylor approximation around $\theta_0$ and assuming convergence in model training, Koh and Liang show that the change in model weight $\Delta\theta_{-\mathbf{z}} = \theta_{-\mathbf{z},\epsilon} - \theta_0$ would be:

$$\Delta\theta_{-\mathbf{z},\epsilon} \approx H^{-1} \nabla_\theta \mathcal{L}(\mathbf{z}, \theta_0)\epsilon. \tag{8}$$

Influence function is then defined by the rate of change of loss on the testing point (or synthesized image) $\hat{\mathbf{z}}$ with respect to the perturbation $\epsilon$ on the training point $\mathbf{z}$. By the chain rule:

$$\frac{\partial \mathcal{L}(\hat{\mathbf{z}}, \theta_{-\mathbf{z},\epsilon})}{\partial \epsilon} = \frac{\partial \mathcal{L}(\hat{\mathbf{z}}, \theta_{-\mathbf{z},\epsilon})}{\partial \theta_{-\mathbf{z},\epsilon}} \frac{\partial \theta_{-\mathbf{z},\epsilon}}{\partial \epsilon} \approx \nabla_\theta \mathcal{L}(\hat{\mathbf{z}}, \theta_0)^T H^{-1} \nabla_\theta \mathcal{L}(\mathbf{z}, \theta_0). \tag{9}$$

To make the computation of Equation 9 tractable, recent works [14, 2, 54, 53, 15, 55] estimate the Hessian by Generalized Gauss-Newton (GGN) approximation, which essentially replaces the Hessian with the Fisher information matrix.

However, even with this approximation, the tradeoff between computation cost and attribution performance remains. The size of the gradient (e.g., $\nabla_\theta \mathcal{L}(\mathbf{z}, \theta_0)$) is too big to store. One could either sacrifice runtime by computing training point gradients on the fly upon each query [54, 55, 1], or projecting gradients to a much smaller dimension that sacrifices performance [14, 2, 15].

### A.3 Eigenvalue-Corrected Kronecker-factored Approximate Curvature (EKFAC)

We collect attribution data using AbU+, an improved variant of AbU [1], by replacing the diagonal Fisher-information approximation with EKFAC [101]. Below is a brief overview of EKFAC. We first discuss KFAC, the basis of EKFAC.

**KFAC** (Kronecker-factored Approximate Curvature) consists of two core ideas to reduce the space complexity of FIM: (1) approximating the Fisher information matrix (FIM) blockwise, where each layer corresponds to a block, and (2) reducing each layer's FIM block to two smaller covariances. To illustrate the idea, for a linear layer (no bias) at layer $i$:

$$\begin{aligned} s_i &= W_i\, a_{i-1}, \\ a_i &= \varphi_i(s_i), \end{aligned} \tag{10}$$

where $\varphi_i$ is an element-wise nonlinearity (e.g., ReLU [120]), $a_{i-1} \in \mathbb{R}^{d_{\text{in}}}$, and $W_i \in \mathbb{R}^{d_{\text{out}} \times d_{\text{in}}}$. Writing $g_i = \nabla_{s_i} \mathcal{L}$, the weight gradient is

$$\begin{aligned} \nabla_{W_i} \mathcal{L} &= g_i\, a_{i-1}^T, \\ \text{vec}(\nabla_{W_i} \mathcal{L}) &= a_{i-1} \otimes g_i, \end{aligned} \tag{11}$$

where $\text{vec}(\cdot)$ flattens a 2D matrix column-wise into an vector. We use the fact that a flattened outer product of two vectors can be written as a Kronecker product of the two vectors. The Fisher block becomes

$$
\begin{aligned}
F_i &= \mathbb{E}\big[\operatorname{vec}(\nabla_{W_i}\mathcal{L})\operatorname{vec}(\nabla_{W_i}\mathcal{L})^T\big] \\
&= \mathbb{E}\big[(a_{i-1}a_{i-1}^T)\otimes(g_i g_i^T)\big] \\
&= \underbrace{\mathbb{E}\big[(a_{i-1}\otimes g_i)(a_{i-1}\otimes g_i)^T\big]}_{d_{\mathrm{in}}d_{\mathrm{out}}\times d_{\mathrm{in}}d_{\mathrm{out}}} \approx \underbrace{\mathbb{E}[a_{i-1}a_{i-1}^T]}_{d_{\mathrm{in}}\times d_{\mathrm{in}}} \otimes \underbrace{\mathbb{E}[g_i g_i^T]}_{d_{\mathrm{out}}\times d_{\mathrm{out}}}.
\end{aligned}
\tag{12}
$$

The last step in the Equation 12 leverages properties of the Kronecker product. The final approximation reduces space complexity from $\mathcal{O}(d_{\mathrm{in}}^2 d_{\mathrm{out}}^2)$ to $\mathcal{O}(d_{\mathrm{in}}^2 + d_{\mathrm{out}}^2)$, since the only requirement is to compute the two smaller covariance matrices.

**EKFAC** refines KFAC by correcting the eigenvalues without increasing space complexity. We first take the eigen decompositions of the two covariance matrices:

$$
A = \mathbb{E}[a_{i-1}a_{i-1}^T] = U_A\, S_A\, U_A^T, \quad B = \mathbb{E}[g_i g_i^T] = U_B\, S_B\, U_B^T.
\tag{13}
$$

Then the FIM block becomes:

$$
\begin{aligned}
F_i = A \otimes B &= (U_A\, S_A\, U_A^T) \otimes (U_B\, S_B\, U_B^T) \\
&= (U_A \otimes U_B)\,(S_A \otimes S_B)\,(U_A \otimes U_B)^T.
\end{aligned}
\tag{14}
$$

EKFAC fixes the shared eigenbasis $U_A \otimes U_B$ but re-estimates the diagonal eigenvalue matrix $S$ by projecting empirical gradients into this basis:

$$
F_i \approx (U_A \otimes U_B)\, S\, (U_A \otimes U_B)^T.
\tag{15}
$$

Since $U_A \in \mathbb{R}^{d_{\mathrm{in}}\times d_{\mathrm{in}}}$, $U_B \in \mathbb{R}^{d_{\mathrm{out}}\times d_{\mathrm{out}}}$, and $S$ is diagonal of size $d_{\mathrm{in}}d_{\mathrm{out}}$, the $\mathcal{O}(d_{\mathrm{in}}^2 + d_{\mathrm{out}}^2)$ cost is preserved while reducing approximation error [101].

In practice, we sample training images, noise vectors, and timesteps to estimate $A$, $B$, and the corrected eigenvalues $S$. Implementation details, including convolutional extensions, appear in Section B.

### A.4 Learning-to-Rank Objectives

In Section 4 of the main text, we compare our cross-entropy objective with two other alternatives: MSE loss and ordinal loss. For the two losses, we follow the conventions from Pobrotyn *et al.* [103].

Using the notations from Section 3 of the main text, we want to predict the normalized ranked scores $\pi_{\hat{\mathbf{z}}}^i \in [\frac{1}{K}, \frac{2}{K}, ..., 1]$ for each training sample $\mathbf{z}_i$, where the most influential sample is assigned $\frac{1}{K}$, and the least assigned 1.

**MSE loss.** This simply regresses the normalized rank:

$$
\mathop{\mathbb{E}}_{\substack{\hat{\mathbf{z}}\sim\hat{\mathcal{Z}} \\ \mathbf{z}_i\sim\hat{\mathcal{D}}_{\hat{\mathbf{z}}}}} \left\| \pi_{\hat{\mathbf{z}}}^i - \sigma_{\alpha,\beta}\big(r_\psi(\hat{\mathbf{z}},\mathbf{z}_i)\big) \right\|^2,
\tag{16}
$$

where $r_\psi$ and the affine-scaled sigmoid $\sigma_{\alpha,\beta}(x) = 1/\big(1 + e^{-(\alpha x+\beta)}\big)$ are as defined in Section 3.2. As reported in Section 4.1, training with this loss fails to converge and yields poor retrieval accuracy. We conjecture that regressing dense ranks over $10^4$ candidates (vs. the usual $5-100$) [60, 103] makes the MSE formulation ill-suited to our setting.

**Ordinal loss.** In the ordinal-regression framework [59], each ground-truth (unnormalized) rank $r \in \{1, \ldots, K\}$ is converted into a binary vector of length $K-1$ via

$$
b^k(r) = \begin{cases} 1, & r > k, \\ 0, & r \leq k, \end{cases} \qquad k = 1, \ldots, K.
\tag{17}
$$

Our ranker $r_\psi$ now outputs $K-1$ logits $\ell^1, \ldots, \ell^K$, which we transform into probabilities

$$
p^k = \sigma(\ell^k) = \frac{1}{1 + e^{-\ell^k}}.
\tag{18}
$$

The ordinal loss is the sum of binary cross-entropies over thresholds:

$$\mathcal{L}_{\text{ord}} = - \mathbb{E}_{\hat{\mathbf{z}}, i} \sum_{k=1}^{K-1} \left[ b^k\left(\pi_{\hat{\mathbf{z}}}^i\right) \log p^k \; + \; \left(1 - b^k(\pi_{\hat{\mathbf{z}}}^i)\right) \log(1 - p^k) \right]. \tag{19}$$

At inference, we recover a scalar rank via

$$\hat{r} \; = \; \sum_{k=1}^{K-1} p^k. \tag{20}$$

Directly using $K \approx 10^4$ would require $10^4$ binary heads and thresholds, which is memory-prohibitive. Instead, we coarsen the rank range into $B = 10$ equal-width bins and apply ordinal loss over the $B - 1$ thresholds, reducing the number of binary classifiers to 9, while preserving most of the ordinal structure.

In contrast to our setup in Section 3 of the main text, the ordinal approach requires the network to emit multiple feature heads (one per threshold), compute a separate cosine similarity, affine transform, and sigmoid for each, and then sum all resulting probabilities to recover a scalar rank. This multi-step, multi-head pipeline increases parameters and computation and slows inference. Since our cross-entropy-based method matches ordinal loss in accuracy while only requiring one feature vector with no extra summation, we adopt it as our main method due to its simplicity and efficiency.

# B  Implementation Details

## B.1  MSCOCO Models

**Collecting attribution data.** We follow the AbU+ procedure from Wang *et al.* [1], performing a single unlearning step that updates only the cross-attention key/value layers. When using EKFAC in place of a diagonal Fisher approximation, we set the Newton step-size to $0.01/N$, with $N = 118\,287$ (the size of the MSCOCO training set).

Within that step, we estimate the diffusion loss via Monte Carlo by sampling and averaging over 50,000 independent (noise, timestep) pairs. Attribution scores are then defined as the difference in loss between the unlearned and original models. To compute each loss, we evaluate at 20 equally spaced timesteps; at each timestep, we average the five losses obtained by combining the image with each of its five corresponding captions (using different random noise for each caption).

**Training rank models.** Our rank model is a 3-layer MLP with hidden and output dimensions of 768. We optimize using AdamW (learning rate $10^{-3}$, default betas $0.9, 0.999$, weight decay $0.01$) for 10 epochs on the training set, without any additional learning-rate scheduling.

## B.2  Stable Diffusion

**Collecting attribution data.** Similar to MSCOCO (Section B.1, we run AbU+ by performing a single unlearning step that updates only the cross-attention key/value layers. We set the Newton step-size to $0.002$, and $N = 400,000,000$ (the size of LAION-400M [20]).

Within the unlearning step, we estimate the diffusion loss via Monte Carlo by sampling and averaging over 4,000 independent (noise, timestep) pairs. To compute the loss for attribution scores, we evaluate at 10 equally spaced timesteps, and at each time step, we sample 1 random noise and use the corresponding caption to assess the loss value.

**Training rank models.** We follow the exact same training recipe as the rank model for MSCOCO, which is described in Section B.1.

## B.3  Baselines

We describe the baselines used in our experiments. Most baselines follow those reported in AbU [1].

**Pixel space.** Following JourneyTRAK's implementation [53], we flatten the pixel intensities and use cosine similarity for attribution.

**CLIP image and text features.** We use the official ViT-B/32 model for image and text features.

**DINO [92].** We use the official ViT-B/16 model for image features.

**DINOv2 [121]** We use the official ViT-L14 model with registers for image features.

**CLIP (AbC) and DINO (AbC) [95].** We use the official models trained on the combination of object-centric and style-centric customized images. CLIP (AbC) and DINO (AbC) are selected because they are the best-performing choices of features.

**TRAK [14] and Journey TRAK [53].** We adopt the official implementation of TRAK and Journey-TRAK and use a random projection dimension of 16384, the same as what they use for MSCOCO experiments.

**D-TRAK [2].** We follow the best-performing hyperparameter reported in D-TRAK, using a random projection dimension of 32768 and lambda of 500. We use a single model to compute the influence score.

**AbU [1] and AbU+.** For AbU, we follow the default hyperparameter reported in the paper. For AbU+, we follow the same hyperparameters as the ones used for data curation, which is described in Section B.1.

**Licenses.** Below we list the licenses of code, data, and models we used for this project.

- **MSCOCO source model:** collected from Georgiev *et al.* [53], which is under the MIT License.
- **MSCOCO dataset:** Creative Commons Attribution 4.0 License.
- **MSCOCO synthesized images testset:** collected from AbU [1], which is under CC BY-NC-SA 4.0.
- **Stable Diffusion:** CreativeML Open RAIL++-M License.
- **DiffusionDB images:** MIT License.
- **CLIP model:** MIT License.
- **DINO model:** Apache 2.0.
- **DINOv2 model:** Apache 2.0.
- **AbC model:** CC BY-NC-SA 4.0.
- **TRAK code:** MIT License.
- **EKFAC code:** Taken from the Kronfluence codebase, which is under Apache 2.0 License.

## C Additional Analysis

### C.1 Compute Cost

Our experiments are all done by NVIDIA A100 GPUs. Below, we describe the runtime cost of the components of our project.

**Data curation.** On MSCOCO, attributing 100k candidates for 110 test queries at 2 hours/query took 220 GPU-hours, while curating 10k candidates for 5,000 train/val queries at 0.25 hour/query took 1,250 GPU-hours. For Stable Diffusion, attributing 20k candidates for 5,050 train/val queries at 3 hours/query required 15,150 GPU-hours, and 140 test queries at 15 hours/query consumed 2,100 GPU-hours in total.

**Training time for LTR models.** Training one rank model on MSCOCO for 10 epochs takes approximately 1 GPU-hour, while training the Stable Diffusion rank model for the same number of epochs requires about 6 GPU-hours.

### C.2 Additional Results

**Effectiveness of K-NN retrieval using off-the-shelf features.** We study the effect of sampling from top-$K$ neighbors for data collection, described in Section 3.2. As in Table 1, we report in Table 2 counterfactual metrics for AbU+ ran on (1) the full training set (**AbU+**) and (2) top neighbors after K-

| Method | Loss deviation $\Delta\mathcal{L}(\hat{\mathbf{z}}, \theta)$ $\uparrow (\times 10^{-3})$ | | | Image deviation $\Delta G_\theta(\epsilon, \mathbf{c})$ MSE $\uparrow (\times 10^{-2})$ | | | CLIP $\downarrow (\times 10^{-1})$ | | |
|---|---|---|---|---|---|---|---|---|---|
| | 500 | 1000 | 4000 | 500 | 1000 | 4000 | 500 | 1000 | 4000 |
| Random | 3.5±0.0 | 3.5±0.0 | 3.5±0.0 | 4.1±0.1 | 4.1±0.1 | 4.0±0.1 | 7.9±0.0 | 7.9±0.0 | 7.9±0.0 |
| D-TRAK | 5.4±0.2 | 6.6±0.2 | 9.6±0.3 | **5.9**±0.2 | **6.4**±0.3 | **7.8**±0.3 | 7.3±0.1 | 7.1±0.1 | 6.4±0.1 |
| AbU+ | **5.8**±0.2 | **7.1**±0.2 | **11.0**±0.3 | 5.6±0.2 | 6.2±0.2 | 7.5±0.2 | 7.2±0.1 | 6.8±0.1 | **5.8**±0.1 |
| AbU+ (K-NN) | **5.8**±0.2 | **7.1**±0.2 | 9.7±0.4 | 5.4±0.2 | 6.3±0.3 | 6.9±0.3 | **7.1**±0.1 | **6.7**±0.1 | **5.8**±0.1 |

Table 2: **Counterfactual leave-$K$-out evaluations.** We report loss deviation and image deviation metrics at $K \in \{500, 1000, 4000\}$. Values are scaled as indicated in the headers; gray shows the standard error.

NN (**AbU+ (K-NN)**). We also copied numbers from D-TRAK (2nd best teacher) and random baselines as reference. We find that applying K-NN retrieval does not introduce a significant performance drop.

**Predicting absolute attribution scores directly.** We explore directly predicting the absolute attribution instead of ranks, where the MLP regresses the absolute attribution scores (normalized by mean and standard deviation). This regression leads to worse ranking performance (0.009 **mAP (1000)**) than our learning-to-rank method (0.724 **mAP (1000)**).

**Rank prediction evaluation.** In Section 4, we only include mAP ($L$) with one $L$ value. Here, we include mAP (500) and mAP (4000) for MSCOCO experiments in Figure 10,11,12,13,14,15. We include mAP (500) and mAP (1000) for Stable Diffusion experiments in Figure 16,17. All trends are similar to the ones reported in the main text.

**MSCOCO qualitative results.** Figure 8 presents additional MSCOCO examples. Our rank model can retrieve influential training images as AbU+. Their influence is confirmed-removing those images, retraining, and regenerating the query leads to large deviations.

**Stable Diffusion qualitative results.** Figure 9 shows additional examples on Stable Diffusion. Our model's top attributions follow those of AbU+, emphasizing prompt content over pure visual similarity. Since full counterfactual retraining is infeasible at this scale, we cannot definitively verify that these attributions reflect true influence. However, as shown in Section 4.2, our method reliably predicts AbU+'s ranks, indicating a consistent attribution signal.

Establishing the ground-truth validity of this signal is left to future work. Prior studies suggest that influence-based methods may weaken on very large models (e.g., LLMs) [122, 123]. However, developing more robust attribution algorithms for large-scale models—and distilling a rank model from stronger teacher methods using our method—are all promising directions ahead.

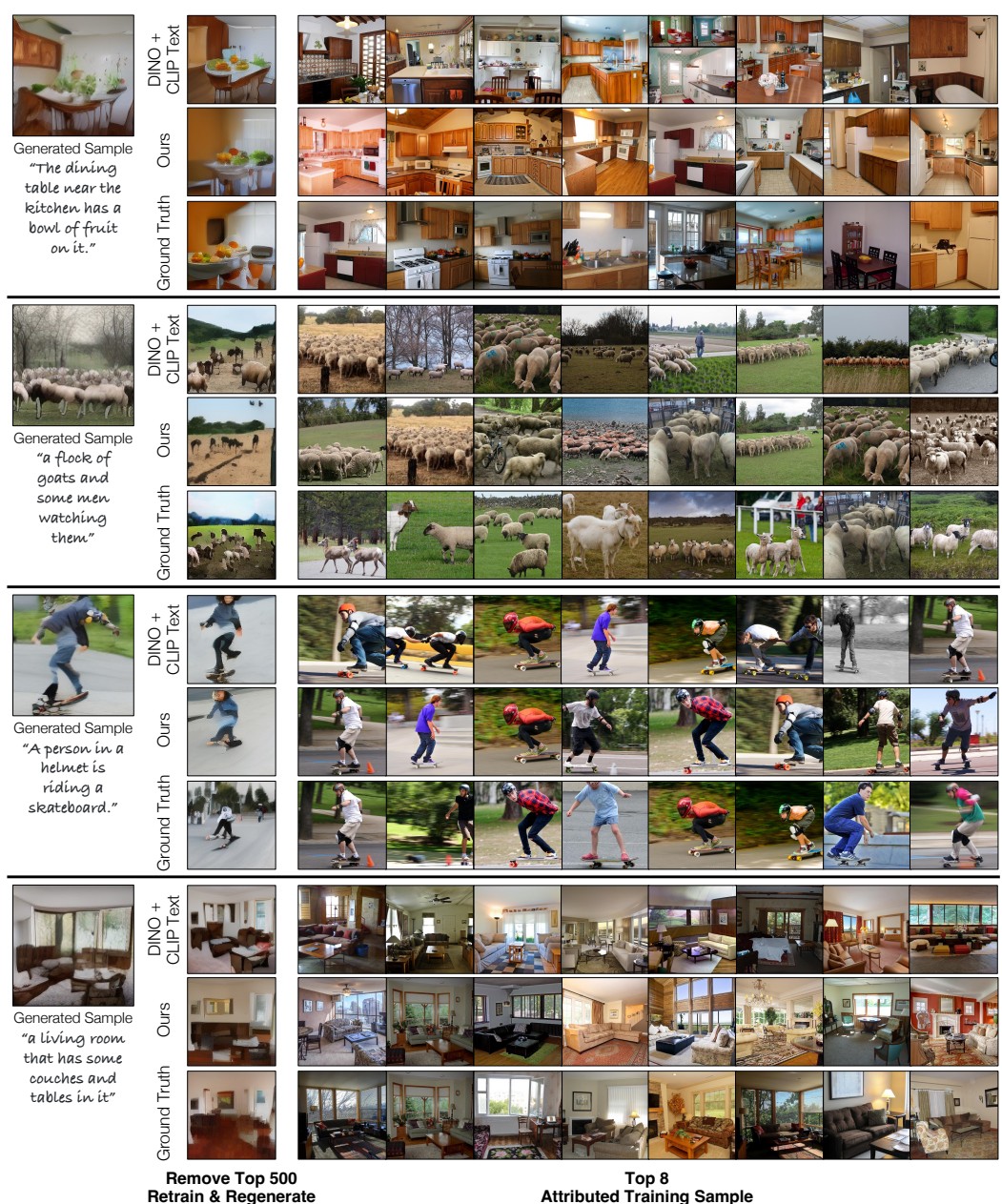

Figure 8: **MSCOCO qualitative results.** For each synthesized sample (leftmost), we compare three methods—DINO+CLIP-Text (top row), our tuned rank model (middle row), and AbU+ ground truth (bottom row). *Left block*: after removing the top 500 attributed images and retraining, the model re-generates the query. *Right block*: each method's top-8 attributed training samples. Our method matches AbU+ in both forgetting behavior and retrieved examples.

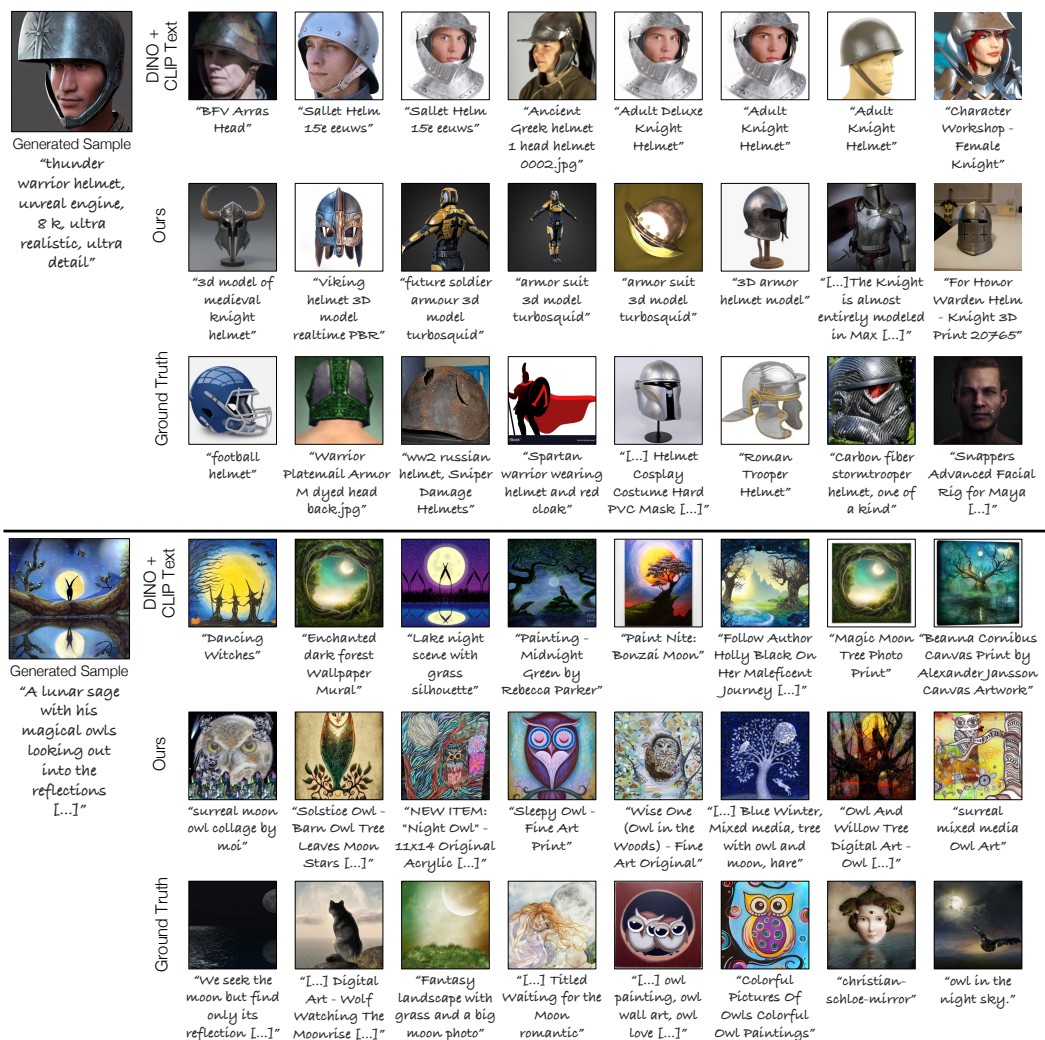

Figure 9: **Stable diffusion qualitative results.** For each generated image (left), we compare the DINO+CLIP-Text baseline (top row), our calibrated feature ranker (middle row), and AbU+ ground-truth attributions (bottom row). Both AbU+ and our method tend to retrieve images that reflect textual cues rather than visual similarity. *Top*: "warrior helmet" – retrieved helmets rather than faces wearing helmets. *Bottom*: "lunar sage . . . magical owls" – retrieved owl-centric scenes despite no owls in the query images.

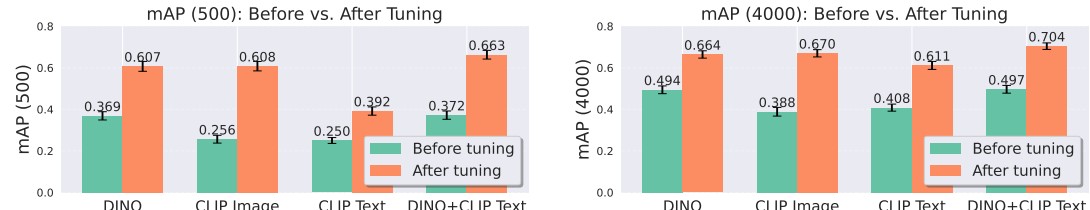

Figure 10: **mAP across different feature spaces.** We compare different feature spaces, before and after tuning for attribution. We report mAP (500) and mAP (4000) to the ground truth ranking, generated by AbU+. The trend is similar to Figure 3 (left) of the main text.

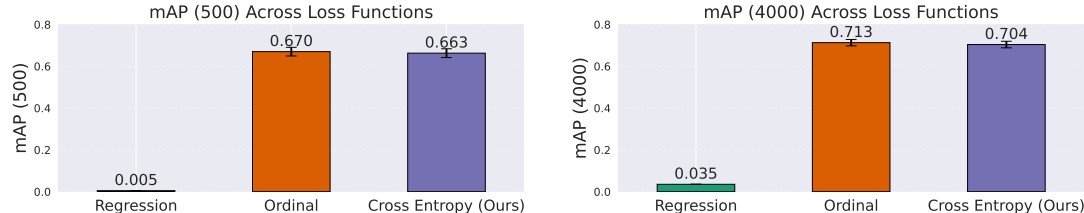

Figure 11: **mAP across different learning-to-rank losses.** Simple MSE regression does not converge well. Our cross-entropy method achieves performance similar to ordinal loss while supporting similarity search. We report mAP (500) and mAP (4000), and the trend is similar to Figure 3 (right) of the main text.

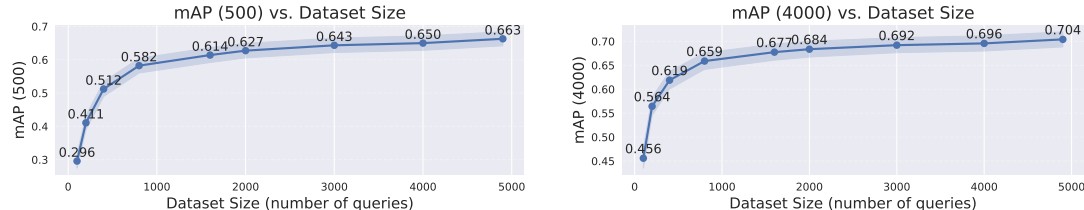

Figure 12: **mAP across different dataset sizes.** We find that the performance quickly improves and saturates as the dataset size grows. We report mAP (500) and mAP (4000), and the trend is similar to Figure 4 (left) of the main text.

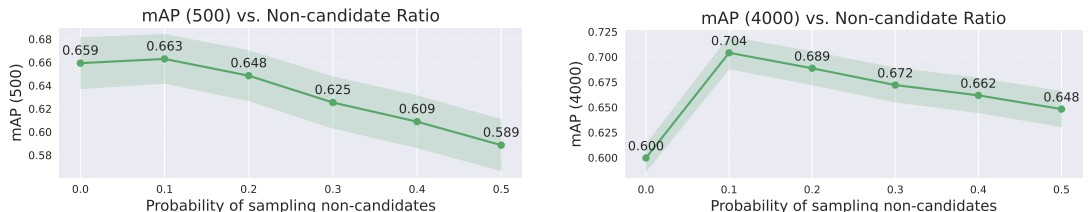

Figure 13: **mAP across probability of non-candidate sampling.** Using a few randomly sampled, unrelated images from the training set helps keep the learned attribution model, while having too many impedes the learning. We report mAP (500) and mAP (4000), and the trend is similar to Figure 4 (right) of the main text.

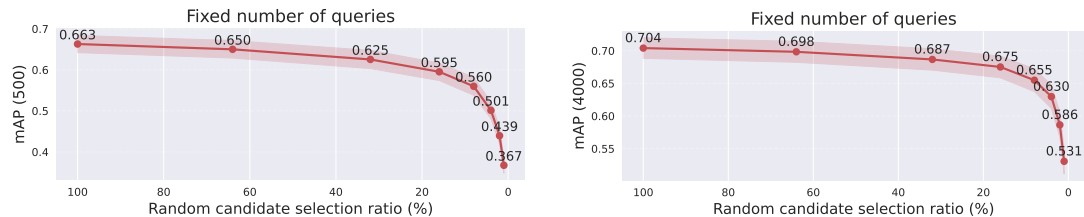

Figure 14: **mAP vs. random subset ratio with a fixed number of queries.** Reliable rankings can be learned, even with relatively fewer training images per query. We report mAP (500) and mAP (4000), and the trend is similar to Figure 5 (left) of the main text.

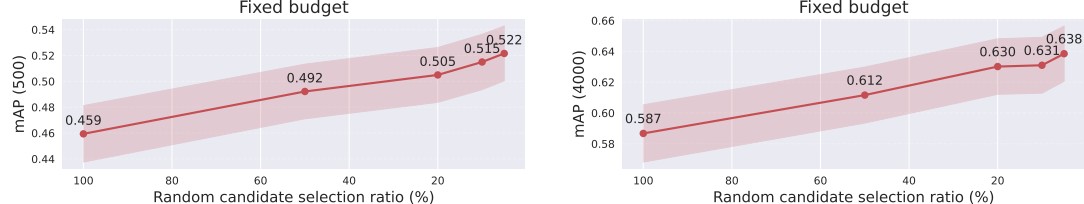

Figure 15: **mAP vs. random subset ratio with a fixed budget.** We find that at a fixed budget of 2.45M, more query images with fewer training images are beneficial. We report mAP (500) and mAP (4000), and the trend is similar to Figure 5 (right) of the main text.

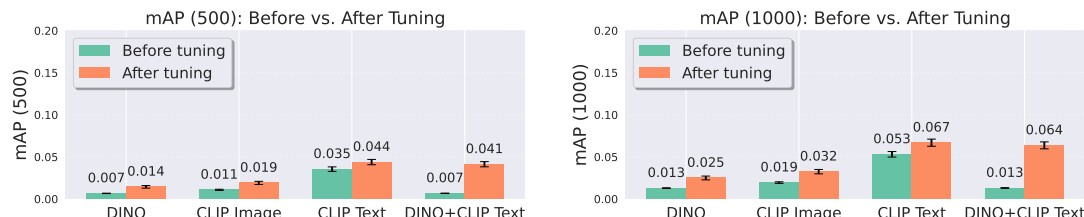

Figure 16: **Stable Diffusion ranking results (tuning feature spaces).** We see similar trends as with MS-COCO, with the strongest performing embedding using both text and image features. However, text features are more necessary to yield strong performance in this setting. We report mAP (500) and mAP (1000), and the trend is similar to Figure 6 (right) of the main text.

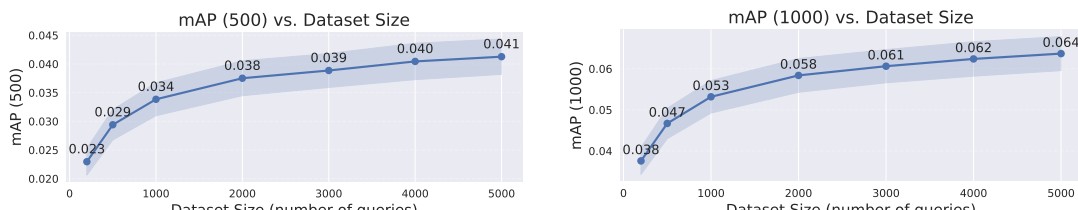

Figure 17: **Stable Diffusion ranking results (dataset sizes).** Performance increases with query images, increasing additional gains with more compute dedicated to gathering attribution training data. We report mAP (500) and mAP (1000), and the trend is similar to Figure 6 (right) of the main text.

