# OpenReview forum: "Fast Data Attribution for Text-to-Image Models"
_NeurIPS.cc/2025/Conference — NeurIPS 2025 poster_

### Official Review · Reviewer_QpB6 · 2025-06-21

**Clarity:** 2
**Significance:** 2
**Originality:** 2
**Rating:** 3
**Confidence:** 3

**Summary:**

This paper proposes a novel approach for scalable and efficient data attribution. The key idea is to distill a slow, unlearning-based attribution method to a feature embedding space for efficient retrieval of highly influential training images. Good experimental results have been reported. The authors claim "our method can achieve better or competitive performance in a few seconds, faster than existing methods by 2,500× ∼ 400,000×."
If this is true, it's impressive. On the other hand, it's not possible to verify here. The author stated at the end of their abstract: "Our code, models, and datasets will be made publicly available."

**Questions:**

The paper reports good performance of their methods; however, the justification is not convincingly presented. The proposed method is not rigorously presented either. See early comments on this. Since the source codes are not made available, it is impossible to validate their results. If the authors want to make a convincing case, they may want to consider releasing their code and the code for their experiments, so that others can reproduce their results.

**Ethical Concerns:**

["NO or VERY MINOR ethics concerns only"]

**Final Justification:**

See my response to the authors.

**Limitations:**

There are some discussions under Section 5 Discussion, Broader Impacts, and Limitations

**Paper Formatting Concerns:**

No.

**Quality:**

2

**Strengths And Weaknesses:**

The paper is okay up to the bottom of page 3. Then it becomes very vague to understand.
For example, in equation (1) at the bottom of page 3, the symbols in the equation are not adequately defined. Readers have to imagine what they are by following some traditions. However, it's the authors' responsibility to declare them.
The following content, from the top of page 4 to the end of Section 3, is presented in a handwaving style without providing a rigorous description.
Perhaps the authors could improve by providing a clear and readable algorithm here.
Without knowing exactly what the method is, it is hard to appreciate its reported success.

This reviewer decides to recommend negatively to this submission, mainly because, based on the submitted work, it is hard to see what the proposed algorithm is. Additionally, since the work is not reproducible, it is impossible to validate the numerical success declared in the abstract, namely "demonstrating that our method can achieve better or competitive performance in a few seconds, faster than existing methods by 2,500× ∼ 400,000×." Considering the paper's main contribution is the declared experimental success, which is not verifiable.

---

> ### Author Rebuttal · Authors · 2025-07-30
>
> Thanks for the helpful suggestions and comments. Below are our responses.
>
> **Code release.** As mentioned in the abstract and in the review, we will release code, data, evaluation script, and models to ensure reproducibility upon publication.
>
> **Writing improvement.** Thank you for suggesting writing improvement for our method. While other reviewers didn’t raise issues with the writing, we agree that it can be polished, and we are happy to provide any further clarifications during the discussion period.
>
> **Symbols in Equation 1 (AbU+).**
> In Eqn 1, $\theta_0$ and $\hat z$ are defined in L138, and $\theta_{-\hat z}$, $\alpha$, and $F$ are defined in L139. $N$ is the size of the training set. $\mathcal L$ is the training loss of text-to-image diffusion model, as defined in Section A.1 in the supplemental. In the revision, we will more clearly refer to Section A.1 and define all the symbols.
>
> **"Clear and readable algorithm."** For reference, we provide a rigorous algorithm block below, which will be added to the revision.
>
>
>
> ---
> *(We use mathjax in this response. Please refresh the page if it's not properly rendered, thanks!)*
> >**Algorithm 1 Data-curation (Sec. 3.1).**
> >
> >For distillation, we curate a teacher set with **AbU+**. For each synthetic query, we first retrieve the top-$K$ neighbors to narrow the search space, then compute AbU+ scores only on that subset (optionally subsampling to size $M$) to improve efficiency. Finally, we rank all scored neighbors for that query and store the ranks. We normalize the rank to $[0, 1]$.
> >
> > **Input**
> > – $ \mathcal D=\{(x_i,c_i)\}_{i=1}^{N} $:&nbsp;&nbsp; Training set (image–caption pairs) \
> > – $ \phi $:&nbsp;&nbsp; Frozen feature encoder  for retrieval  \
> > – $ \tau(\hat z,z) $:&nbsp;&nbsp; Attribution scores from slow teacher AbU+  \
> > – $ \mathcal C $:&nbsp;&nbsp; Caption set  to generate query images  \
> > – $ K $:&nbsp;&nbsp; Retrieval cut-off \
> > – $ M $:&nbsp;&nbsp; Subsample size
> >
> >  **Algorithm**
> > 1. Initialize teacher attribution set $ \mathcal U \leftarrow \varnothing $.
> > 2. **For each** caption $ c\in\mathcal C $:
> > 3. &nbsp;&nbsp;&nbsp;&nbsp;Generate synthetic image $ \hat x $ and form query $ \hat z=(\hat x,c) $.
> > 4. &nbsp;&nbsp;&nbsp;&nbsp;Retrieve top-$K$ neighbors $ \mathcal D_{\hat z}\subset\mathcal D $ of $ \hat z $ using $ \phi $.
> > 5. &nbsp;&nbsp;&nbsp;&nbsp;Uniformly subsample $ \tilde{\mathcal D_{\hat z}}\subset\mathcal D_{\hat z} $ with $ |\tilde{\mathcal D_{\hat z}}| = M $.
> > 6. &nbsp;&nbsp;&nbsp;&nbsp;Initialize list $ \mathcal S_{\hat z}\leftarrow\varnothing $.
> > 7. &nbsp;&nbsp;&nbsp;&nbsp;**For each** $ z_i\in\tilde{\mathcal D_{\hat z}} $:
> > 8. &nbsp;&nbsp;&nbsp;&nbsp;&nbsp;&nbsp;&nbsp;&nbsp;Compute score $ s_i=\tau(\hat z,z_i) $ and append $s_i$ to $ \mathcal S_{\hat z} $.
> > 9. &nbsp;&nbsp;&nbsp;&nbsp;Sort $ \mathcal S_{\hat z} $ in **descending** order by $ s_i $ to assign ranks $ \pi^i_{\hat z}=1,2,\dots,|\mathcal S_{\hat z}| $ to each training point $z\in\tilde{\mathcal D_{\hat z}}$.
> >10. &nbsp;&nbsp;&nbsp;&nbsp;**For each** $ z_i\in\tilde{\mathcal D_{\hat z}} $ and its rank $\pi^i_{\hat z}$:
> >11. &nbsp;&nbsp;&nbsp;&nbsp;&nbsp;&nbsp;&nbsp;&nbsp;Normalize and update $ \pi^i_{\hat z} \leftarrow \dfrac{\pi^i_{\hat z}}{|\mathcal S_{\hat z}|} $.
> >12. &nbsp;&nbsp;&nbsp;&nbsp;&nbsp;&nbsp;&nbsp;&nbsp;Append $ (\hat z,z_i, \pi^i_{\hat z},\mathcal D_{\hat z}) $ to $ \mathcal U $.
> >
> > **Output** Curated teacher set $ \mathcal U $ containing tuples $ (\hat z,z_i, \pi^i_{\hat z},\mathcal D_{\hat z})$.
> >
> >(We introduce a new symbol $\mathcal U$ for clarity, and other symbols follows the same convention in the submission.)
>
>
> ---
>
> >**Algorithm 2 Training the fast attribution embedder  (Sec. 3.2).**
> >
> >Using the teacher set $\mathcal U$, we finetune features so that the feature similarity between the training and synthesized image predicts the ranks.
> >
> > **Input**
> > – $ \mathcal D$:&nbsp;&nbsp; Training set   (from Algorithm 1) \
> > – $ \mathcal U $:&nbsp;&nbsp; Teacher set  (from Algorithm 1) \
> > – $ \phi $:&nbsp;&nbsp; Frozen encoder \
> > – $ g_\psi $:&nbsp;&nbsp; Trainable MLP \
> > – $ (\alpha,\beta) $:&nbsp;&nbsp; Trainable logit scale \
> > – $ p_n $:&nbsp;&nbsp; Probability of sampling non-neighbor images *(for learning to assign lower rank to unrelated image)* \
> > – $ \eta $:&nbsp;&nbsp; Learning rate \
> > – $T$ :&nbsp;&nbsp; Training iteration
> >
> > **Algorithm**
> > 1. **Repeat** for $T$ iterations:
> > 2. &nbsp;&nbsp;&nbsp;&nbsp;Sample $ (\hat z, z_i, \pi^i_{\hat z},\mathcal D_{\hat z}) $ from $ \mathcal U $.
> > 3. &nbsp;&nbsp;&nbsp;&nbsp; `Sample non-neighbor` With probability $ p_n $: \
> > &nbsp;&nbsp;&nbsp;&nbsp;&nbsp;&nbsp;&nbsp;&nbsp;  (a) `replace data with a non-neighbor` $z_i \leftarrow z \sim \mathcal D /\mathcal D_{\hat z}$ \
> > &nbsp;&nbsp;&nbsp;&nbsp;&nbsp;&nbsp;&nbsp;&nbsp;  (b) `set the lowest rank to the non-neighbor` $\pi^i_{\hat z} \leftarrow 1$
> > 4. &nbsp;&nbsp;&nbsp;&nbsp;`Get features` $f_\psi(\hat z) = g_\psi\bigl(\phi(\hat z)\bigr),\quad f_\psi(z_i) = g_\psi\bigl(\phi(z_i)\bigr) $.
> > 5. &nbsp;&nbsp;&nbsp;&nbsp;`Similarity` $ r_\psi(\hat z, z_i) = \cos\bigl(f_\psi(\hat z), f_\psi(z_i)\bigr) $.
> > 6. &nbsp;&nbsp;&nbsp;&nbsp;`Predicted rank`  $ \sigma_{\alpha,\beta}(r_\psi(\hat z, z_i)) = \operatorname{sigmoid}(\alpha\odot r_\psi(\hat z,z_i) + \beta) $.
> > 7. &nbsp;&nbsp;&nbsp;&nbsp;`BCE Loss`  $ \mathcal L(\psi, \alpha, \beta) = \ell_\text{BCE}\bigl(\pi^i_{\hat z}\ ,\ \sigma_{\alpha,\beta}(r_\psi(\hat z, z_i))\bigr) $.
> > 8. &nbsp;&nbsp;&nbsp;&nbsp;Update $ \psi,\alpha,\beta $ using $ \nabla\mathcal L $ with AdamW optimizer with learning rate $ \eta $.
> > 9. `Store the optimized parameters` $ \psi^\* \leftarrow \psi$, $ \alpha^\* \leftarrow \alpha$, $ \beta^\* \leftarrow \beta$.
> >
> > **Output** Trained $ \psi^\*, \alpha^\*, \beta^\* $; pre-compute and index $ f_{\psi^\*}(z_i) $ for all training images $z_i \in \mathcal D$.
> >
> >(Besides $\mathcal U$, we introduce $p_n$, $T$ for clarity, and other symbols follow the same convention in the submission.)

---

> > ### Comment · Reviewer_QpB6 · 2025-08-01
> >
> > This reviewer appreciates the authors' inclusion of algorithmic details in the rebuttal. However, this reviewer remains surprised that none of the other reviewers appear to share the concern articulated in the original review regarding the lack of a convincing justification: "The paper is okay up to the bottom of page 3. Then it becomes very vague to understand. For example, in equation (1) at the bottom of page 3, the symbols in the equation are not adequately defined. Readers have to imagine what they are by following some traditions. However, it's the authors' responsibility to declare them. The following content, from the top of page 4 to the end of Section 3, is presented in a handwaving style without providing a rigorous description. Perhaps the authors could improve by providing a clear and readable algorithm here. Without knowing exactly what the method is, it is hard to appreciate its reported success".
> >
> > While the additional algorithmic information is noted, it does not fully address the issues raised above. That said, after considering the perspectives of the other reviewers, this reviewer is willing to raise the score from 2 to 3.

---

### Official Review · Reviewer_viaY · 2025-06-26

**Clarity:** 3
**Significance:** 3
**Originality:** 2
**Rating:** 5
**Confidence:** 5

**Summary:**

This paper presents a method for efficiently identifying the training images that most influenced a generated image from a text-to-image model.
The primary challenge with existing data attribution techniques is their high computational cost, which makes them impractical for large-scale use.
The authors' key contribution is an approach that significantly speeds up data attribution.
They achieve this by "distilling" the knowledge from a slow but relatively accurate attribution method, AbU+, into a lightweight feature embedding model.
Once trained, this new model can quickly find influential training images by performing a fast similarity search in its learned feature space, eliminating the need to run the expensive attribution algorithm for every new query.
The paper demonstrates through extensive experiments on both the MSCOCO and the large-scale LAION datasets that their method achieves attribution performance comparable to or better than existing methods, but is thousands of times faster.

**Questions:**

See weaknesses

**Ethical Concerns:**

["NO or VERY MINOR ethics concerns only"]

**Final Justification:**

The response has effectively addressed my primary concerns.

**Limitations:**

yes

**Quality:**

3

**Strengths And Weaknesses:**

Strengths
1. The paper addresses a highly relevant and timely problem.
As generative models become more widespread, methods for tracing generated content back to training data are crucial for addressing issues of copyright, artist compensation, and model transparency.
Existing attribution methods are often too computationally expensive for practical deployment.
This work significantly contributes by dramatically reducing the runtime of attribution to a few milliseconds, making it faster.

2. The authors validate their method on MSCOCO using counterfactual evaluation, which is the "gold standard" for attribution.
This involves removing the top-k identified training images, retraining the model, and measuring the degradation in its ability to produce the query image.
This is a costly but powerful way to demonstrate true influence.
The authors perform a systematic study of various design choices, providing clear justifications for their final model. This includes comparing different feature spaces to tune (e.g., DINO, CLIP Text, and their combination), evaluating multiple learning-to-rank objectives (MSE, Ordinal, Cross-Entropy), and analyzing the impact of data-scaling and sampling strategies.

3. The successful application to Stable Diffusion/LAION demonstrates the method's scalability and its ability to handle modern, large-scale systems.

4. The paper is well-written and easy to follow. The methodology is presented clearly, with helpful illustrations like Figure 2 that effectively communicate the overall pipeline.

Weaknesses
1. The core idea of distilling a slow data attribution method into a fast, feature-based model might not be novel. Previous papers have explored similar paradigms, and the authors did not discuss them [1, 2].

2. The LTR approach distills the relative ranking of influential examples but loses the raw, calibrated attribution scores produced by the teacher. This means that while the method can identify which images were most influential, it cannot quantify how much more influential one image is than another or determine if influence is concentrated in a few examples or diffuse among many.

3. The authors did not adopt the widely used Linear Datamodeling Scores as on of their evaluation metrics, making it a bit difficult to compare with previous methods.

[1] MATES: Model-Aware Data Selection for Efficient Pretraining with Data Influence Models. NeurIPS 2024

[2] Data Selection via Optimal Control for Language Models. ICLR 2025

---

> ### Author Rebuttal · Authors · 2025-07-30
>
> Thank you for acknowledging that our paper addresses a timely problem, provides a systematic study of design choices, demonstrates a scalable method, and presents clear methodology. We address your comments below.
>
> ---
> **Comparison to previous works.**
> Thanks for the suggestion! We will discuss them in the revision. There are several differences between their approaches and ours:
>
> - **The task is different.** They focus on data selection for a specified downstream task, where they assign a single influence score to each training point and select ones with high influence. We focus on training point influence for *any given synthesized image*. The training point influence is different per query synthesized image.
>
> - **The distillation method is different.** They take the training point feature as input and regress the single influence score. We use feature similarity between the training and synthesized image to predict influence, where we finetune features using learning-to-rank objectives to encourage similarities to have the correct rank.
>
> - **Modality is different.** They focus on language models, and we focus on text-to-image models.
>
> ---
> **"The LTR approach distills the relative ranking."**
> Yes, we agree and have discussed this in the limitation (L330). Following Reviewer PCEG’s suggestion, we conducted a preliminary experiment on regressing the absolute attribution scores (normalized by mean and standard deviation). This regression leads to worse ranking performance than our learning-to-rank method. Below, we report mAP (1000) for comparison (higher is better):
>
> | Regress Absolute Score  ||| Ours |
> |:---:|--|--|:---:|
> | 0.009 ± 0.000 ||| **0.724 ± 0.021** |
>
>
> Predicting accurate absolute attribution scores remains challenging, and it will be interesting to explore improvements in this in the future. We will add this discussion in the revision.
>
> ---
> **Linear Datamodeling Scores (LDS).**
> LDS was introduced by Park et al. [1], with the assumption that attribution methods are additive, where the influence of a group of training points is the sum of the individual training point influences. As noted by AbU [2], while the additive assumption sometimes holds true (e.g., influence functions), it does not hold true for many methods (e.g., AbU, feature similarity).
>
> Meanwhile, we did compare previous methods with the counterfactual forgetting metric, as used by many prior works [1,2,3,4]. We will clarify this in the revision.
>
> [1] Park et al. TRAK: Attributing Model Behavior at Scale.
>
> [2] Wang et al. Data Attribution for Text-to-Image Models by Unlearning Synthesized Images.
>
> [3] Ilyas et al. Datamodels: Predicting Predictions from Training Data.
>
> [4] Georgiev et al. The Journey, Not the Destination: How Data Guides Diffusion Models.

---

> > ### Comment · Reviewer_viaY · 2025-08-04
> >
> > Thank you to the authors for their thorough and thoughtful rebuttal.
> >
> > Your response has effectively addressed my primary concerns. The clarification regarding the key differences between your work and prior art—specifically in the task definition, distillation method, and modality—is well-articulated and convincing.
> >
> > I particularly appreciate the new preliminary experiment comparing your learning-to-rank (LTR) method with the regression of absolute scores. The significant difference in performance (mAP of 0.724 vs. 0.009) provides strong empirical justification for your design choice and substantially strengthens the paper. The justification for using the counterfactual forgetting metric in place of Linear Datamodeling Scores is also reasonable.
> >
> > Given these clarifications and the proposed additions to the manuscript, I am pleased to raise my score to 5. I look forward to seeing the revised version.

---

### Official Review · Reviewer_sTkC · 2025-06-30

**Clarity:** 2
**Significance:** 3
**Originality:** 3
**Rating:** 4
**Confidence:** 3

**Summary:**

This paper proposes a scalable and efficient data attribution method for text-to-image generative models. The key idea is to distill an unlearning-based attribution method into a task-specific embedding space, enabling fast retrieval of influential training images via similarity search. The learned embedding is trained with a learning-to-rank objective over attribution scores generated by AbU+. The method is benchmarked on both MSCOCO and Stable Diffusion, demonstrating superior attribution accuracy while achieving high speedup over compared attribution methods.

**Questions:**

1. How does the teacher model limit the overall performance?
2. Why are these two metrics selected, and are others infeasible due to the problem setting?

**Ethical Concerns:**

["NO or VERY MINOR ethics concerns only"]

**Final Justification:**

The paper proposes a novel and practical attribution method with solid empirical results. While it depends on AbU+ as the teacher and does not recover absolute scores, the design is well-motivated and the evaluation is thorough. Despite some limitations, the contribution is clear and valuable.

**Limitations:**

Yes.

**Paper Formatting Concerns:**

None.

**Quality:**

3

**Strengths And Weaknesses:**

**Strengthens**
1. The idea of learning an embedding space specifically optimized for data attribution is novel. It bridges the gap between accurate attribution and similarity-based methods.
2. The proposed method achieves faster attribution compared to unlearning or influence-function-based methods, making it practical for deployment in real-world applications.
3. The experiments are comprehensive. They include ablations on feature types, loss functions and sampling strategies.

**Weaknesses**
1. The proposed method is fully supervised by AbU+ scores. However, the paper does not analyze whether the teacher model itself is stable or accurate across modalities. Any failure case in AbU+ would be learned.
2. While the method achieves fast inference, it sacrifices the ability to interpret absolute attribution scores (only rankings are learned). Furthermore, constructing the training set still requires significant computational cost.
3. Both evaluation metrics (ranking prediction accuracy and counterfactual forgetting) are aligned with the design of this paper. It remains unclear whether these metrics correlate well with real attribution quality.

---

> ### Author Rebuttal · Authors · 2025-07-30
>
> Thank you for acknowledging that our paper proposes a novel idea, offers a practical attribution application, and conducts comprehensive experiments. We address your comments below.
>
> ---
> **"Any failure case in AbU+ would be learned."**
> Yes, we agree and have discussed this in the limitation (L333). This would be true for all distillation methods. Like all distillation methods, our method can be generally applied to different teacher methods, enabling future improvements in (slow) teachers to be distilled into fast students, using our procedure.
>
> ---
> **Predicting relative instead of absolute attribution scores.**
> Yes, we agree and have discussed this in the limitation (L330). Following Reviewer PCEG’s suggestion, we conducted a preliminary experiment on regressing the absolute attribution scores (normalized by mean and standard deviation). This regression leads to worse ranking performance than our learning-to-rank method. Below, we report mAP (1000) for comparison (higher is better):
>
> | Regress Absolute Score  ||| Ours |
> |:---:|--|--|:---:|
> | 0.009 ± 0.000 ||| **0.724 ± 0.021** |
>
> Predicting accurate absolute attribution scores remains challenging, and it will be interesting to explore improvements in this in the future. We will add this discussion in the revision.
>
> ---
> **Computational cost of the training set construction.**
> Yes, the cost of data curation remains high, and we will discuss this in the limitations. We will clarify that our work focuses on achieving efficient attribution, with a low storage cost, **at inference time**. Currently, most attribution methods cannot handle user requests in real time, making it difficult to deploy in real-world use cases. In scenarios where an attribution system handles many daily user requests, our method is more favorable, since it provides a cost-efficient and fast solution at inference time.
>
> ---
> **"Do these metrics correlate well with real attribution quality?"**
> Yes, the counterfactual forgetting metric directly measures the real attribution quality, which is widely adopted by AbU and other recent works [1,2,3,4]. As described in AbU [1], the metric evaluates attribution in the following way:
>
> >If an attribution algorithm accurately identifies influential training images for the generated image $\hat{\textbf{z}}$, then a model trained without those images would be incapable of generating or representing $\hat{\textbf{z}}$.
>
> Meanwhile, as discussed in L227-231, we use cheap rank prediction metrics (e.g., mAP (1000)) to evaluate design choices for ranking models. To reduce computation cost, we only perform expensive counterfactual metrics on the best-ranking models to confirm that our method indeed yields good attribution quality. We will clarify this in the revision. We are happy to consider other metrics suggested by the reviewer.
>
> [1] Wang et al. Data Attribution for Text-to-Image Models by Unlearning Synthesized Images.
>
> [2] Ilyas et al. Datamodels: Predicting Predictions from Training Data.
>
> [3] Park et al. TRAK: Attributing Model Behavior at Scale.
>
> [4] Georgiev et al. The Journey, Not the Destination: How Data Guides Diffusion Models.

---

> > ### Comment · Reviewer_sTkC · 2025-08-01
> >
> > Thank you for the response and additional experiments. The comparison with absolute score regression and the clarification on counterfactual forgetting are appreciated. The method remains dependent on the teacher model, which limits its robustness. However, this is a common limitation, and the paper reasonably discusses its extensibility. I will keep my score unchanged.

---

### Official Review · Reviewer_PCEG · 2025-07-02

**Clarity:** 3
**Significance:** 4
**Originality:** 4
**Rating:** 5
**Confidence:** 4

**Summary:**

This paper addresses the computational bottleneck in data attribution for text-to-image models by proposing a learning-to-rank approach that distills slow but accurate attribution methods into fast feature embeddings. The authors leverage Attribution by Unlearning (AbU) as a teacher method and train a feature space that can identify influential training images through simple cosine similarity search. Their method achievessignificant speedup compared to existing approaches while maintaining competitive attribution performance.

**Questions:**

- Have you explored modifications to preserve absolute attribution scores rather than just rankings? This could be achieved through regression with appropriate normalization or by learning a calibration function. This information could be valuable for understanding whether influence is concentrated in a few images or distributed across many.

- Also wondering if ensembling multiple different teacher ()attribution) models may achieve better results?

**Ethical Concerns:**

["NO or VERY MINOR ethics concerns only"]

**Final Justification:**

The paper tackles a critical challenge where existing attribution methods have oversized cost (compute/time), making real-world use infeasible. The proposed speedup makes attribution practical. The rebuttal also quelled many weakness and questions i had.

**Limitations:**

- This method can only work if one has access to the pretrained T2I model
- One additional limitation, is the computational cost of the data curation phase, which still requires running expensive attribution methods

**Paper Formatting Concerns:**

Nil

**Quality:**

3

**Strengths And Weaknesses:**

streghts first:
- The paper tackles a critical challenge where existing attribution methods have oversized cost (compute/time), making real-world use infeasible. The proposed speedup makes attribution practical.
- The two-stage data strategy (coarse retrieval followed by fine-grained ranking) balances computational efficiency with attribution accuracy. The learning-to-rank framework is appropriate for distilling ranking information from expensive teacher methods.
- The evaluation is thorough, including both efficient rank prediction metrics and expensive counterfactual evaluation. The ablation studies investigate key design choices (feature spaces, loss functions, sampling strategies) with clear insights.
- Finally the method achieves attribution performance comparable to influence-function and Unlearning-based methods while operating at millisecond latencies with minimal storage overhead.

Weaknesses:
- The approach is inherently limited by the teacher method's quality. Any biases, errors, or failure modes in AbU+ can be inherited, limiting the path to improve attribution beyond the teacher.
- This is minor, nonetheless, from the perspective of training data creators, by learning only relative rankings rather than absolute attribution scores, the method loses potentially important information about influence magnitude and concentration. This could matter for applications like fair compensation where the degree of influence is critical. But that in itself would require massive scale and doesn't undermine the efficacy of this method or the results.
- The initial k-NN retrieval using off-the-shelf features may miss influential images that are semantically related but visually dissimilar, creating an irrecoverable performance ceiling.

---

> ### Author Rebuttal · Authors · 2025-07-30
>
> Thank you for acknowledging that our paper addresses a critical challenge, proposes an appropriate learning-to-rank framework, offers a thorough evaluation, and achieves low-latency attribution with minimal storage requirements. We address your comments below.
>
> ---
> **"The approach is inherently limited by the teacher method's quality."**
> Yes, we agree and have discussed this in the limitation (L333). This would be true for all distillation methods. Like all distillation methods, our method can be generally applied to different teacher methods, enabling future improvements in (slow) teachers to be distilled into fast students, using our procedure.
>
> ---
> **"Ensembling multiple different teacher (attribution) models may achieve better results?"**
> Thank you for your suggestion. We also believe that if an ensemble leads to better attribution rankings, we expect our method to perform better by distilling from those rankings. We plan to explore it in future work.
>
> ---
> **Predicting relative instead of absolute attribution scores.**
> Yes, we agree and have discussed this in the limitation (L330). Following your suggestion, we conducted a preliminary experiment on regressing the absolute attribution scores (normalized by mean and standard deviation). This regression leads to worse ranking performance than our learning-to-rank method. Below, we report mAP (1000) for comparison (higher is better):
>
> | Regress Absolute Score  ||| Ours |
> |:---:|--|--|:---:|
> | 0.009 ± 0.000 ||| **0.724 ± 0.021** |
>
>
> Predicting accurate absolute attribution scores remains challenging, and it will be interesting to explore improvements in this in the future. We will add this discussion in the revision.
>
> ---
> **Effectiveness of K-NN retrieval using off-the-shelf features.**
> Following the reviewer’s suggestion, we study the effect of applying K-NN for data collection. As in Table 1 in the main text, we report counterfactual metrics for AbU+ ran on (1) the full training set (**AbU+**) and (2) top neighbors after K-NN (**AbU+ (K-NN)**). We also copied numbers from D-TRAK (2nd best teacher) and random baselines as reference:
>
> |  | Loss Change $\uparrow$ [K=500] ($\times 10^{-3}$) | Loss Change $\uparrow$ [K=1000] ($\times 10^{-3}$) | Loss Change $\uparrow$ [K=4000] ($\times 10^{-3}$) | MSE $\uparrow$ [K=500] ($\times 10^{-2}$) | MSE $\uparrow$ [K=1000] ($\times 10^{-2}$) | MSE $\uparrow$ [K=4000] ($\times 10^{-2}$) | CLIP $\downarrow$ [K=500] ($\times 10^{-1}$) | CLIP $\downarrow$ [K=1000] ($\times 10^{-1}$) | CLIP $\downarrow$ [K=4000] ($\times 10^{-1}$) |
> |---|:---:|:---:|:---:|:---:|:---:|:---:|:---:|:---:|:---:|
> | Random                | 3.5±0.0 | 3.5±0.0 | 3.5±0.0 | 4.1±0.1 | 4.1±0.1 | 4.0±0.1 | 7.9±0.0 | 7.9±0.0 | 7.9±0.0 |
> | D‑TRAK                | 5.4±0.2 | 6.6±0.2 | 9.6±0.3 | **5.9±0.2** | **6.4±0.3** | **7.8±0.3** | 7.3±0.1 | 7.1±0.1 | 6.4±0.1 |
> | AbU+                  | **5.8±0.2** | **7.1±0.2** | **11.0±0.3** | 5.6±0.2 | 6.2±0.2 | 7.5±0.2 | 7.2±0.1 | 6.8±0.1 | **5.8±0.1** |
> | AbU+ (K-NN)             | **5.8±0.2** | **7.1±0.2** | 9.7±0.4 | 5.4±0.2 | 6.3±0.3 | 6.9±0.3 | **7.1±0.1** | **6.7±0.1** | **5.8±0.1** |
>
>
> We find that applying K-NN retrieval does not introduce a significant performance drop. We are happy to include this study in the revision.
>
> ---
> **Access to the pretrained T2I model.**
> Yes, our method assumes full access to the model, dataset, and training loss. This setup directly applies to a company running attribution on its in-house model. We also note that recent works on data attribution all follow the same setup [1,2,3].
>
> [1] Georgiev et al. The Journey, Not the Destination: How Data Guides Diffusion Models.
>
> [2] Zheng et al. Intriguing Properties of Data Attribution on Diffusion Models.
>
> [3] Wang et al. Data Attribution for Text-to-Image Models by Unlearning Synthesized Images.
>
> ---
> **Computational cost of the data curation phase.**
> Yes, the cost of data curation remains high, and we will discuss this in the limitations. We will clarify that our work focuses on achieving efficient attribution, with a low storage cost, **at inference time**. Currently, most attribution methods cannot handle user requests in real time, making it difficult to deploy in real-world use cases. In scenarios where an attribution system handles many daily user requests, our method is more favorable, since it provides a cost-efficient and fast solution at inference time.

---

> > ### Comment · Reviewer_PCEG · 2025-08-04
> >
> > Thanks for the response and clarifying experiments. I appreciate the regression and K-NN experiments. I will keep my score unchanged.

---

> ### Comment · Area_Chair_tCpu · 2025-08-01
>
> Hello, a gentle reminder for discussions. Authors have provided an extensive rebuttal, and would be great if you can acknowledge it and comment on it!

---

### Decision · Program_Chairs · 2025-09-17

**Decision:**

Accept (poster)

**Comment:**

All reviewers are in agreement that the proposed method of distilling attribution methods into a feature embedding. There was concern about the presentation and the justification, but given that multiple reviewers recommend the presentation to be a strength of the paper, which the AC's quick read also aligns with, the AC recommends accepting the paper.